# Viruses hijack FPN1 to disrupt iron withholding and suppress host defense

Li Tong[1,4], Jie Wang[1,4], Yunjin Ma[1], Chunying Wang[1], Yue Fu[1], Qi Li[1], Chengjiang Gao ●[2], Hui Song[1], Ying Qin ●[1], Chunyuan Zhao[1,3] ✉ & Wei Zhao ●[1] ✉

Viruses rely on intracellular materials, including iron, to complete their life cycles and iron withholding may limit viral infections. However, the mechanisms through which viruses disrupt host iron homeostasis and the impact of intracellular iron on the host's antiviral defense aren't well studied. Here we show that viral infections facilitate the polyubiquitination and degradation of ferroportin (FPN1, the only cellular iron exporter) by upregulating the host E3 ubiquitin ligase DTX3L, leading to an elevation in cellular iron levels. Excessive ferrous iron suppresses type I IFN responses and autophagy by promoting TBK1 hydroxylation and STING carbonylation in macrophages. FPN1 deficiency suppresses host antiviral defense and facilitates viral replication in vitro and in vivo, while DTX3L deficiency has the opposite effect. These results reveal that viruses hijack host FPN1 to disrupt iron withholding and achieve immune escape, and suggest that iron homeostasis maintained by FPN1 is required for the optimal activation of TBK1- and STING-dependent antiviral responses.

The innate immune response serves as the first-line host defense and plays a crucial role in controlling viral infection[1]. Viral DNA or RNA released during replication is sensed by pattern recognition receptors (PRRs), including retinoic acid-inducible gene-I (RIG-I)-like receptors (RLRs) and cyclic guanosine monophosphate (GMP)-AMP (cGAMP) synthase (cGAS). These innate immune sensors trigger different signaling cascades to initiate type I IFN responses, inflammation, and autophagy, which are critical for viral elimination. RLRs sense single- or double-stranded RNA generated during viral infection to activate the adaptor protein mitochondrial antiviral signaling (MAVS)[2]. Conversely, cGAS recognizes viral dsDNA and generates cGAMP, which in turn binds to and activates the key adaptor stimulator of interferon genes (STING)[3–5]. Both STING and MAVS subsequently activate TANK-binding kinase 1 (TBK1) and IRF3 to initiate the expression of type I IFNs (including IFN-α and IFN-β) and NF-κB-dependent proinflammatory cytokines (such as TNF-α and IL-6). Type I IFN signaling, in turn, induces the transcription of numerous IFN-stimulated genes (ISGs) through the JAK-STAT pathway, bolstering antiviral responses[1].

Simultaneously, the activation of TBK1 and STING induces autophagy to directly degrade viral particles or components[6–8]. Therefore, TBK1- and STING-mediated type I IFN response and autophagy emerge as critical antiviral defense mechanisms.

Iron is an essential nutrient for both vertebrate hosts and microbial invaders such as bacteria and fungi. Iron serves as a redox catalyst and is involved in various cellular processes[9]. However, excessive iron catalyzes the Fenton reaction, generating hydroxyl radicals that damage lipids, proteins, and DNA. Therefore, cellular iron levels should be tightly controlled to prevent detrimental effects. In addition to overseeing systemic iron metabolism, cellular iron homeostasis is mainly maintained by the iron importer transferrin receptor (TFR), divalent metal ion transporter (DMT1), and the only known cellular iron exporter ferroportin (FPN1)[10]. Bacteria extensively utilize iron as a cofactor in enzymes to catalyze redox reactions, playing a pivotal role in fundamental cellular processes such as respiration, DNA synthesis, and protection from reactive oxygen species[11]. Therefore, the hosts adopt a defense mechanism known as iron withholding to limit iron

[1]Department of Pathogenic Biology, Key Laboratory of Infection and Immunity of Shandong Province, and Key Laboratory for Experimental Teratology of the Chinese Ministry of Education, School of Basic Medical Science, Cheeloo College of Medicine, Shandong University, Jinan, Shandong, China. [2]Department of Immunology, School of Basic Medical Science, Cheeloo College of Medicine, Shandong University, Jinan, Shandong, China. [3]Department of Cell Biology, School of Basic Medical Science, Cheeloo College of Medicine, Shandong University, Jinan, Shandong, China. [4]These authors contributed equally: Li Tong, Jie Wang. ✉e-mail: cyzhao@sdu.edu.cn; wzhao@sdu.edu.cn

acquisition by invading pathogens[12]. Microbes have evolved various countermeasures to disrupt host iron withholding and ensure their survival[13–15]. Bacteria and fungi produce different siderophores and high-affinity iron chelators that selectively scavenge iron from the host iron-binding proteins. Some bacteria express specific membrane receptors or hemophore proteins that target host iron-binding proteins, such as heme, hemopexin, hemoglobin, and lactoferrin, facilitating the sequestration of iron[16]. The intracellular pathogen *Candida glabrata* employs the NRF2-BACH1-Fe homeostasis control system to enhance iron accumulation in phagolysosomes[15]. Unlike other microorganisms, viruses do not possess a cellular structure, relying entirely on host materials for replication and packaging. Despite emerging evidence indicating the beneficial aspects of iron withholding in limiting viral infection[17,18], the precise mechanisms through which viruses disrupt host iron homeostasis and the impact of intracellular iron on host antiviral defenses remain unknown.

As the only known cellular iron exporter, FPN1 plays a central role in intracellular iron homeostasis, so the FPN1 levels need to be tightly regulated. Hepcidin induces the endocytosis and ubiquitination of FPN1, followed by degradation predominantly in lysosomes[19,20]. E3 ubiquitin ligase RNF217 mediates the ubiquitination and degradation of FPN1[21]. In this study, we show that viruses induce the expression of E3 ubiquitin ligase DTX3L, to facilitate the ubiquitination and proteasomal degradation of FPN1 in macrophages. Consequently, viral infection contributes to an increase in cellular iron accumulation; in turn, ferrous iron attenuates type I IFN responses and autophagy by promoting TBK1 hydroxylation and STING carbonylation. FPN1 deficiency suppresses host defense against viruses and facilitates viral replication both in vitro and in vivo, while DTX3L deficiency yields the opposite effect. These results highlight the role of viruses in hijacking FPN1 and DTX3L to disrupt cellular iron withholding, thereby achieving immune escape. Additionally, the study suggests that maintaining iron homeostasis through FPN1 is crucial for the optimal activation of TBK1- and STING-dependent antiviral responses.

## Results

### Viruses disrupt iron withholding via downregulating host FPN1

To explore the impact of viral infections on host intracellular iron homeostasis, we examined the iron levels in virus-infected macrophages by conducting a quenchable iron pool (QIP) analysis[15] (Fig. S1a, b). Herpes simplex virus 1 (HSV-1, a type of DNA virus) and vesicular stomatitis virus (VSV, a type of RNA virus) both decreased the QIP in mouse primary peritoneal macrophages (PMs), signifying an increase in total iron levels in macrophages upon viral infection (Figs. 1a, b, S1b, and S1c). Similarly, macrophages in the spleen and liver of HSV-1- and VSV-infected mice exhibited elevated cellular iron levels (Figs. S1d–g, S9a, b). Following viral invasion, immune cells including macrophages secrete type I IFNs to suppress viral replication. However, no difference was observed in the cellular iron levels in IFN-β-stimulated PMs (Fig. 1c). In macrophages, iron is predominately stored in ferritins, while minute amounts of iron are also present in a free and metabolically active form[22]. There were no changes of ferritin levels after both HSV-1 and VSV infection in PMs and THP-1 cells (Fig. S1h, i). Notably, HSV-1 and VSV infection, but not IFN-β stimulation, promoted the accumulation of intracellular ferrous iron in PMs (Figs. 1d, S1j S9c, e). These findings indicate that viral infection induces the ferrous iron accumulation in the cells.

To clarify the mechanism by which viruses hijack host iron withholding, we initially examined the expression of receptors involved in maintaining cellular iron homeostasis. Our findings revealed that only the sole cellular iron exporter, FPN1, but not the cellular iron importers DMT1, TFR1 or TFR2, downregulated after viral infection in mouse PMs and human THP-1 cells (Figs. 1e, f, S2a, b). Moreover, the membrane FPN1 levels on PMs, but not DMT1, TFR1 or TFR2, were reduced after viral infection (Figs. 1g, h and S2c–e). Consistently, the decreased

membrane FPN1 levels was also observed on the spleen and liver macrophages after HSV-1 and VSV infection (Fig. S2f). Notably, IFN-β has no effect on inducing the production of FPN1, DMT1, TFR1 or TFR2 (Figs. 1i, S2c–e and S9f, g). *Ifnar1* deficiency had no impact on HSV-1 and VSV infection caused decrease of FPN1 expression in PMs (Fig. 1j, k). Moreover, FPN1 expression was not affected by NF-κB inhibitors (JSH-23 and QNZ) nor JAK1 inhibitor itacitinib after viral infection (Fig. S2g–i). These data indicated that viruses, rather than host type I IFN responses, induced a reduction in FPN1 expression.

To further investigate the physiological function of FPN1 (encoded by *Slc40a1*) in controlling iron homeostasis during viral infection, *Slc40a1*^fl/fl mice were crossed with *Lyz2*^cre mice to specifically knockout FPN1 in myeloid cells (referred to as "FPN1^CKO") (Fig. S2j, k). FPN1 deficiency abolished the increase in total cellular iron and ferrous iron levels caused by HSV-1 and VSV infections (Figs. 1l, m, S2l, m, and S9c, d). Taken together, these results strongly suggest that viruses disrupt iron withholding by suppressing the host FPN1 expression. Viruses disrupt iron release, by suppressing the host FPN1 expression

### Viruses induce FPN1 degradation via upregulating DTX3L

HSV-1 and VSV infection accelerated the protein degradation of FPN1 in cycloheximide (CHX, a protein synthesis inhibitor) chase experiments (Fig. 2a, b). Notably, this degradation process was reversed by the proteasome inhibitor MG132, with no effect when treated with the lysosome inhibitors chloroquine and 3-methyladenine(3-MA) (Fig. 2c). These results suggest that viruses promote FPN1 degradation through proteasomal pathway. Ubiquitination is required for the subsequent proteasomal degradation of proteins, and E3 ubiquitin ligases determine substrate specificity. To identify potential E3 ubiquitin ligases that mediate the ubiquitin-proteasome degradation of FPN1, we conducted an RNA-seq assay to analyze the genes upregulated following viral infection. Ten E3 ubiquitin ligases were potently induced in PMs infected with HSV-1, VSV, and Sendai virus (SeV, a type of RNA virus) (Fig. 2d). Among these E3 ubiquitin ligases, TRIM30 (a, b, c, and d), RNF213, and RNF135 were found to be antiviral factors that inhibit viral replication, thus not aligning with our objective[23–25]. We then focused on TRIM21, DTX3L, MARCH5, and RNF114, and examined their interaction with FPN1. Only DTX3L exhibited a robust interaction with FPN1 and substantially promoted FPN1 degradation in human embryonic kidney (HEK) 293 T cells (Figs. 2e and S3a).

To establish the role of DTX3L in FPN1 degradation, we constructed *Dtx3l*-deficient mice (Fig. S3b–d). *Dtx3l* deficiency enhanced the FPN1 expression in HSV-1 and VSV infected PMs (Fig. 2f). The HSV-1 and VSV infection-induced degradation of endogenous FPN1 was abolished in *Dtx3l*-deficient PMs (Fig. 2g and S3e). Consistently, DTX3L overexpression inhibited FPN1 expression, while the DTX3L point mutation (C561AC564A) in the RING domain, substituting cysteine residues with alanine at positions 561 and 564, lost the ability to degrade FPN1 (Fig. 2h). Next, we confirmed that viral infection induced DTX3L expression at both the mRNA and protein levels in macrophages (Fig. S3f, g). NF-κB inhibitors (JSH-23 and QNZ), JAK1 inhibitor itacitinib, and *Ifnar1* deficiency blocked viral infection-induced DTX3L expression (Fig. S3h–p), suggesting that viruses activate NF-κB and IFNAR-JAK to upregulate DTX3L. Collectively, these data indicate that viruses induce DTX3L expression to promote the proteasomal degradation of FPN1.

Polyubiquitination is a key step in the ubiquitin-proteasome degradation pathway. We investigated whether DTX3L mediated FPN1 ubiquitination. Endogenous FPN1 interacted with DTX3L in the resting and virus-infected PMs (Fig. 2i, j). In vitro binding assays further demonstrated a direct interaction between FPN1 and DTX3L (Fig. 2k). Concordantly, the confocal analysis showed the colocalization of FPN1 and DTX3L (Fig. 2l). FPN1 polyubiquitination was markedly enhanced in the presence of DTX3L, but not with DTX3L C561AC564A mutant

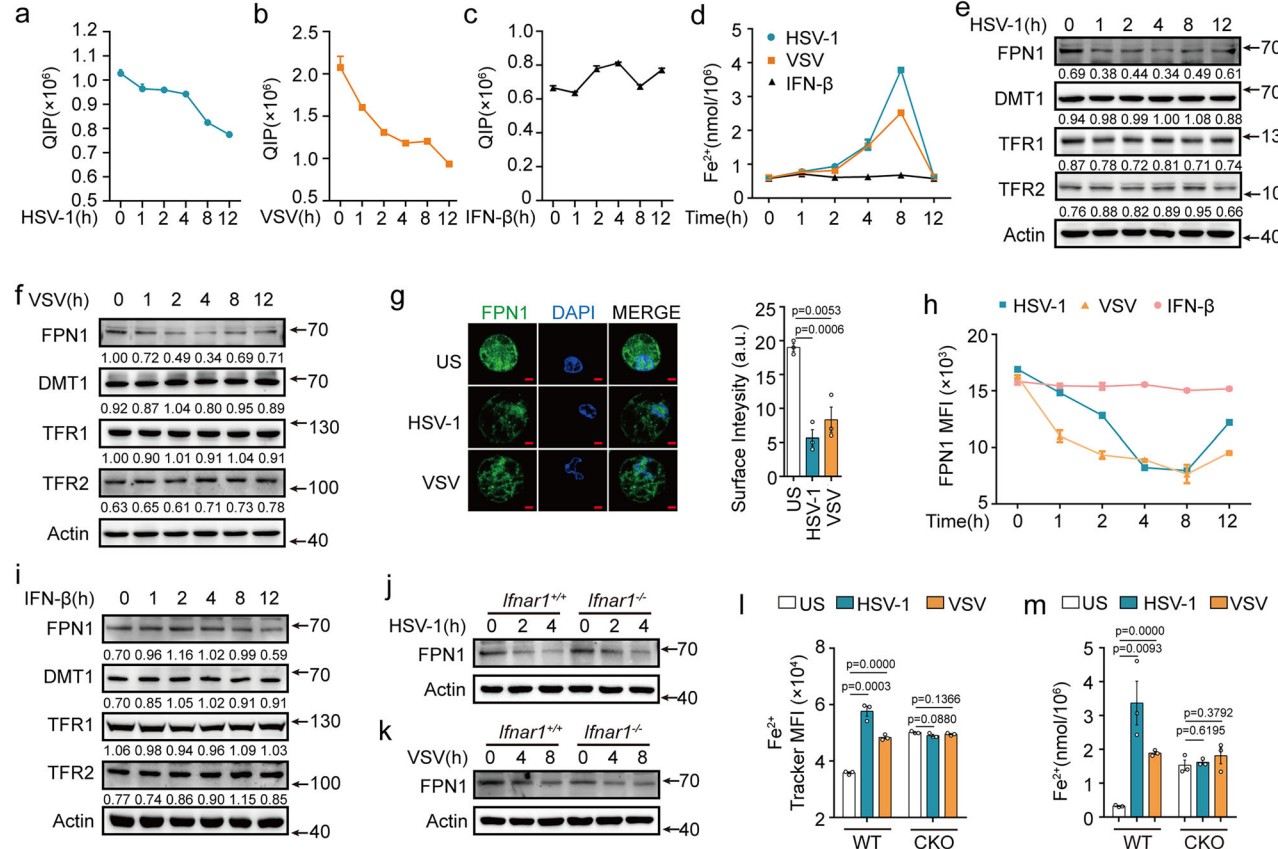

**Fig. 1 | Viruses disrupt iron withholding via downregulating host FPN1. a–c** QIP analysis of intracellular free iron in mouse PMs infected with HSV-1 (**a**), VSV (**b**), or IFN-β (10 ng/mL) (**c**) for the indicated time periods. **d** $Fe^{2+}$ iron detection through iron assay kit in mouse PMs infected with HSV-1, VSV, or IFN-β (10 ng/mL) for the indicated time periods. **e, f** Western blot analysis in mouse PMs infected with HSV-1 (**e**) or VSV (**f**) for the indicated time periods. **g** Confocal analysis of the FPN1-FITC expression on cell surface in mouse PMs infected with HSV-1 or VSV (left). The fluorescence intensity of FPN1 on cell surface was calculated using the "ImageJ" software from three different areas (right), $n = 3$. Scale bars, 3 μm. **h** Flow cytometry analysis of the median fluorescence intensity (MFI) of FPN1-FITC on cell surface in mouse PMs infected with HSV-1, VSV or stimulated with IFN-β (10 ng/mL) for the indicated time periods. Each MFI of FPN1-FITC was calculated using FlowJo_v10,

$n = 3$. The flow cytometry strategy is in Fig. S9f. **i–k** Western blot analysis in mouse PMs stimulated with IFN-β (10 ng/mL) (**i**), or in $Ifnar1^{+/+}$ or $Ifnar1^{-/-}$ PMs infected with HSV-1 (**j**) or VSV (**k**) for the indicated time periods. **l, m** Flow cytometry analysis of $Fe^{2+}$ Tracker (**l**) or $Fe^{2+}$ iron detection through iron assay kit (**m**) of wild-type (WT) and $Slc40a1^{CKO}$ mouse PMs infected with HSV-1 or VSV. The flow cytometry strategy is in Fig. S9c, d. Data are expressed as the mean ± SEM in **a–d, g, h, l**, and **m**. Statistical analyses were performed using two-tailed unpaired Student's t-test. Results were obtained from three independent experiments. US, unstimulated. CKO, $Slc40a1^{CKO}$. Unless otherwise specified, cells infection with HSV-1 (MOI:10) for 4 h, VSV (MOI:1) for 8 h, SeV (MOI:1) for 8 h. Lysates marker sizes in kDa are indicated on the western blot right. Source data are provided as a Source Data file.

and other E3 ligases (including TRIM21, MARCH5, and RNF114), in HEK293T cells (Figs. 2m and S3q). DTX3L selectively enhanced K48-linked, but not K63-linked, polyubiquitination of FPN1 (Fig. 2n, o). Moreover, HSV-1 and VSV infection-induced FPN1 polyubiquitination was completely abolished in *Dtx3l*-deficient PMs (Fig. 2p). Collectively, these data indicate that DTX3L selectively promotes the K48-linked polyubiquitination of FPN1 to induce FPN1 degradation during viral infection.

**Ferrous iron attenuates type I IFN responses and autophagy**
Viral infection led to an accumulation of ferrous iron in macrophages. We investigated the effects of ferrous iron on viral replication and host defense. Ferrous iron facilitated the replication of HSV-1 and VSV, without diminishing the cell viability of mouse PMs (Figs. 3a, b, and S4a–c). This observation suggests that the enhanced cellular iron level favors viral proliferation. Type I IFN response and autophagy are important host defense mechanisms against viral infections. We initially examined the type I IFN pathways and found that ferrous iron dose dependently attenuated the HSV-1 and VSV infection-induced IFN-β expression and phosphorylation of IRF3, TBK1, and STAT1 in PMs (Figs. 3c–f and S4d–f). Similar inhibitory effects were evident in SeV-

infected PMs (Fig. S4f–h). Furthermore, ferrous iron inhibited the HSV-1 and VSV infection-induced expression of IFN-α; regulated upon activation, normal T cell expressed and secreted (RANTES, also known as CCL5); myxovirus resistance protein 1(Mx1); and proinflammatory cytokines (including TNF-α and IL-6) (Fig. S4i–l). Deferoxamine (DFO, an iron chelator) treatment promoted IFN-β expression and enhanced the phosphorylation of IRF3, TBK1, and STAT1 in both HSV-1 and VSV infected PMs (Figs. 3g–i and S4m). In human THP-1 cells, ferrous iron potently attenuated the viral infection-induced expression of IFN-β and the activation of IRF3, TBK1, and STAT1, resulting in the enhancement of viral replication (Fig. 3j–n). Furthermore, ferrous iron also inhibited the phosphorylation of IRF3, TBK1, and STAT1, and the expression of IFN-β induced by interferon-stimulating DNA (ISD, a synthetic dsDNA sequence recognized by cGAS) and polyinosinic: polycytidylic acid [poly(I:C), a synthetic dsRNA sequence recognized by RLRs] (Fig. 3o–q). These results indicate that ferrous iron inhibits RLR- and cGAS-mediated antiviral innate immune responses. Given the crucial role of autophagy in viral clearance, we examined whether ferrous iron affects the autophagy pathway. Ferrous iron treatment decreased the expression of LC3-II in HSV-1 and VSV infected PMs, while DFO treatment enhanced its expression (Fig. 3r–u), indicating

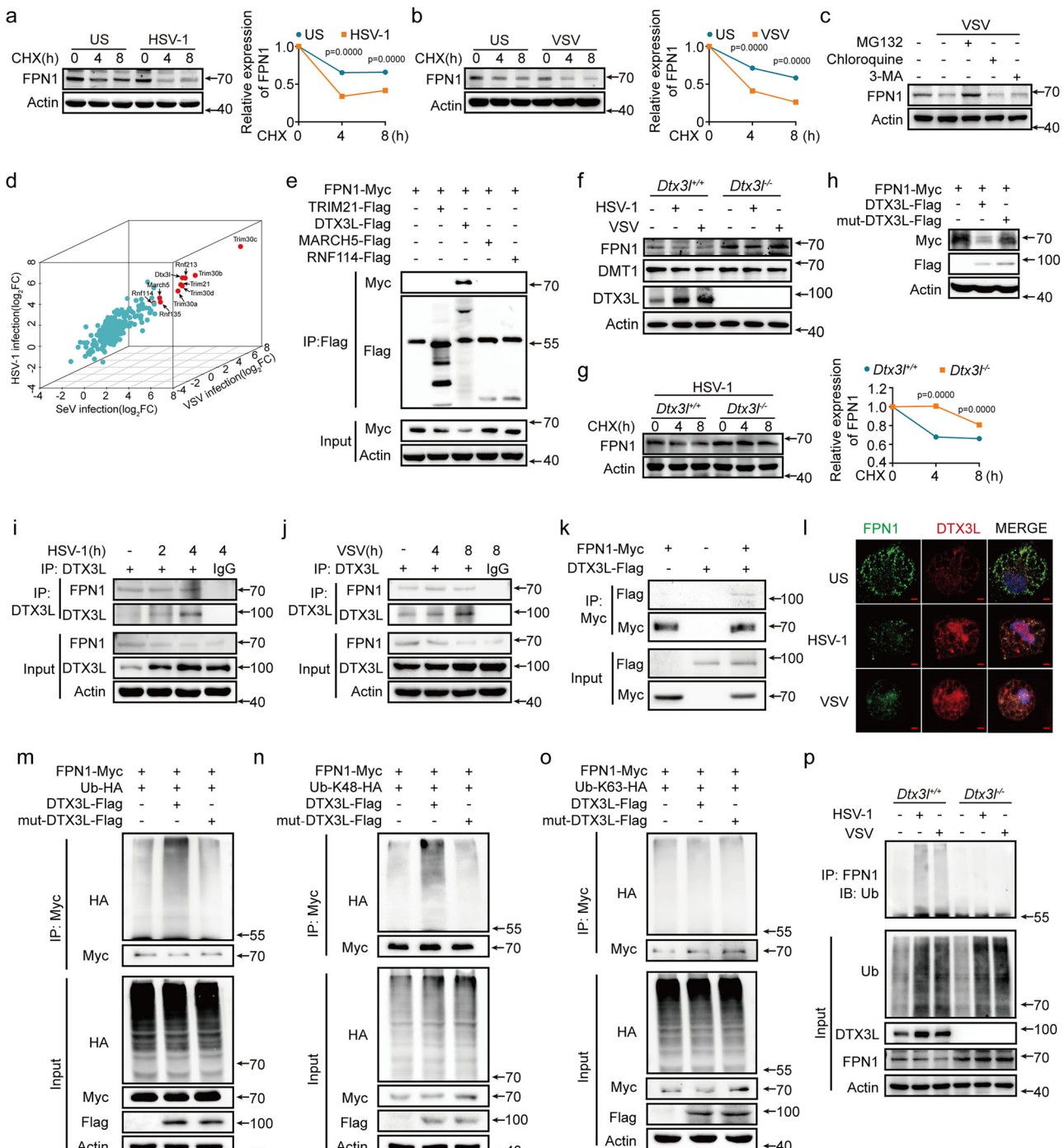

**Fig. 2 | Viruses induce FPN1 degradation via upregulating DTX3L. a, b** Western blot analysis from mouse PMs infected with HSV-1 (**a**) or VSV (**b**) and then treated with cycloheximide (CHX, 10 μM) (left). FPN1 expression level was quantitated by measuring band intensities using the "ImageJ" software (right). **c** Western blot analysis from mouse PMs infected with VSV and then treated with MG132 (10 μM), chloroquine (100 μM), or 3-MA (5 mM) for 4 h. **d** Scatter plot of RNA-sequencing data displays the mouse PMs upregulated E3 ubiquitin ligases in control or infection with SeV, VSV, or HSV-1 (*n* = 3). Colors represent fold-change levels above the values in control (log₂FC≥2: red). (***FDR < 0.001). **e** Immunoprecipitation analysis of the interaction between E3 ligases and FPN1 in HEK293T cells transfected with the indicated plasmids. **f, g** Western blot analysis in *Dtx3l*⁺/⁺ or *Dtx3l*⁻/⁻ PMs infected with HSV-1 or VSV (**f**), or together with CHX (10 μM) (**g**). FPN1 expression level was quantitated by measuring band intensities using "ImageJ" software. **h** Western blot analysis from HEK293T cells transfected with indicated plasmids. **i, j** Western blot

analysis of mouse PMs infected with HSV-1 (**i**) or VSV (**j**), followed by immunoprecipitation with DTX3L antibody. **k** In vitro analysis of the interaction of DTX3L with FPN1. **l** Confocal microscopy of the colocalization between FPN1 and DTX3L in macrophages after infection with HSV-1 or VSV. Scale bars, 2 μm.
**m–o** Immunoprecipitation analysis of the ubiquitination of FPN1 in HEK293T cells transfected with Ub-HA (**m**), Ub-K48-HA (**n**) or Ub-K63-HA (**o**), and other indicated plasmids, with MG132 (10 μM) treatment for 4 h. **p** Lysates from *Dtx3l*⁺/⁺ or *Dtx3l*⁻/⁻ PMs infected with HSV-1 or VSV were immunoprecipitated with FPN1 antibody, followed by western blot analysis. Data are expressed as the mean ± SEM in **a, b,** and **g**. Statistical analyses were performed using two-tailed unpaired Student's t-test. Results were obtained from three independent experiments. US, unstimulated. Unless otherwise specified, cells infection with HSV-1 (MOI:10) for 4 h, VSV (MOI:1) for 8 h, SeV (MOI:1) for 8 h. Lysates marker sizes in kDa are indicated on the western blot right. Source data are provided as a Source Data file.

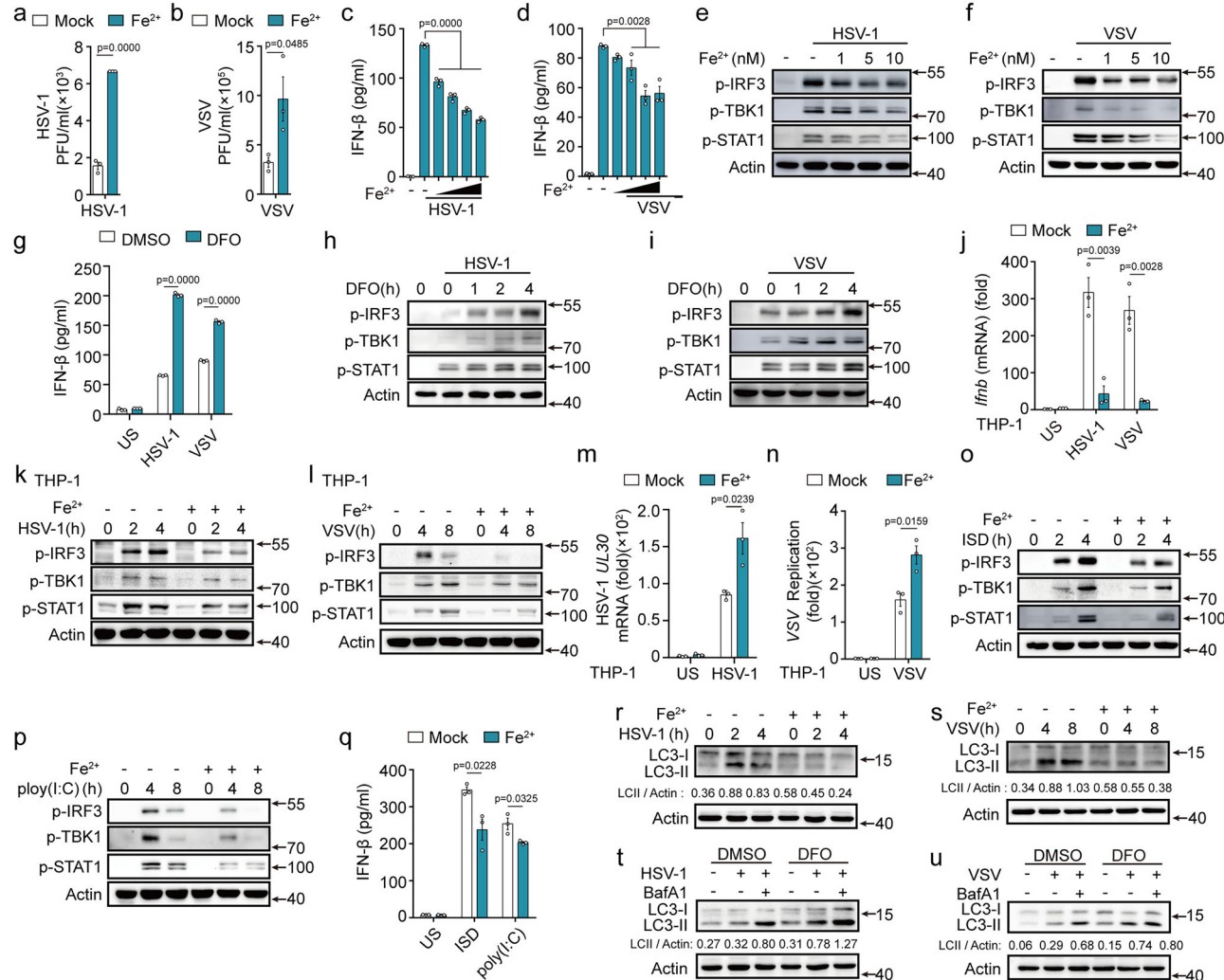

**Fig. 3 | Ferrous iron attenuates type I IFN responses and autophagy to facilitate viral replication. a, b** Viral titers in PMs pretreated with 10 nM Fe$^{2+}$ for 1 h, followed by infection with HSV-1 or VSV. **c–f** ELISA analysis of IFN-β secretion (**c, d**) and western blot analysis (**e, f**) in mouse PMs pretreated with increasing amounts of Fe$^{2+}$ (1, 5, 10, and 20 nM) for 1 h, followed by infection with HSV-1 or VSV. **g** ELISA analysis of IFN-β secretion (**g**), or western blot analysis (**h, i**) in mouse PMs pretreated with deferoxamine (DFO, 10 μM) for 4 h, followed by infection with HSV-1 or VSV. **j–l** RT-PCR analysis of the mRNA expression of *Ifnb* (**j**), or western blot assays (**k, l**) in THP-1 cells pretreated with LAL water (Mock) or Fe$^{2+}$ (10 nM) for 1 h, followed by infection with HSV-1 or VSV. **m, n** RT-PCR analysis of the *HSV-1 UL30* mRNA (**m**) or the replication of VSV(**n**) in THP-1 cells pretreated with Fe$^{2+}$(10 nM) for 1 h, followed by infection with HSV-1 for 8 h or VSV for 12 h. **o–q** Western blot assays (**o, p**), or ELISA analysis of IFN-β secretion (**q**) in PMs pretreated with Fe$^{2+}$ (10 nM) for 1 h, and then transfected with ISD (10 μg/mL) for 4 h or poly(I:C) (10 μg/mL) for 8 h. **r–u** Western blot assays of LC3-II and LC3-I in mouse PMs pretreated or Fe$^{2+}$ (10 nM) for 1 h (**r, s**), and pretreated with BafA1 (100 nM) for 2 h and then treated with DFO (10 μM) for 4 h (**t, u**), followed by infection with HSV-1 (**r, t**) or VSV (**s, u**). LC3-II/actin level was quantitated by measuring band intensities using the "ImageJ" software. Data are expressed as the mean ± SEM in **a–d, g, j, m, n,** and **q**. Statistical analyses were performed using two-tailed unpaired Student's t-test. Results were obtained from three independent experiments. US, unstimulated. Unless otherwise specified, cells infection with HSV-1 (MOI:10) for 4 h, VSV (MOI:1) for 8 h. Lysates marker sizes in kDa are indicated on the western blot right. Source data are provided as a Source Data file.

that ferrous iron treatment attenuated viral infection-induced autophagy.

## Ferrous iron targets TBK1 and STING

To clarify the mechanisms by which ferrous iron suppresses host antiviral defenses, we investigated luciferase promoter activation induced by various receptors and adaptors in innate immune pathways. Ferrous iron treatment significantly reduced the IFN-β and IRF3 promoter activation induced by RIG-I, MDA5, cGAS-STING, MAVS, TRIF, and TBK1 (Fig. 4a, b). However, no difference in the IRF3-induced IFN-β promoter activation was observed in ferrous iron-treated cells (Fig. 4a). Therefore, we speculated that ferrous iron targeted upstream of IRF3. Excessive iron induces the formation of harmful hydroxyl radicals (OH). TBK1, the key kinase that induces

IRF3 activation, can be hydroxylated at proline (P) 48 to inactivate its kinase activity[26]. Given the potent hydroxyl radical-generating ability of ferrous iron, we examined its impact on TBK1 hydroxylation. As expected, ferrous iron promoted TBK1 hydroxylation in both HSV-1 and VSV infected PMs (Fig. 4c, d). In addition, ferrous iron enhanced TBK1 hydroxylation in wild-type TBK1 transfected mouse embryonic fibroblasts (MEFs) and HEK-293T cells, while no hydroxylation was observed in TBK1 P48A mutant (with substitution of the proline residue with alanine at position 48)-transfected cells (Figs. 4e and S5a). Interestingly, ferrous iron had no effects on the expression of IFN-β and phosphorylation of IRF3 and STAT1 induced by TLR9 (Fig. S5b, c), a pathway independent of TBK1. These findings indicate that ferrous iron specifically targets TBK1 to suppress innate immune response.

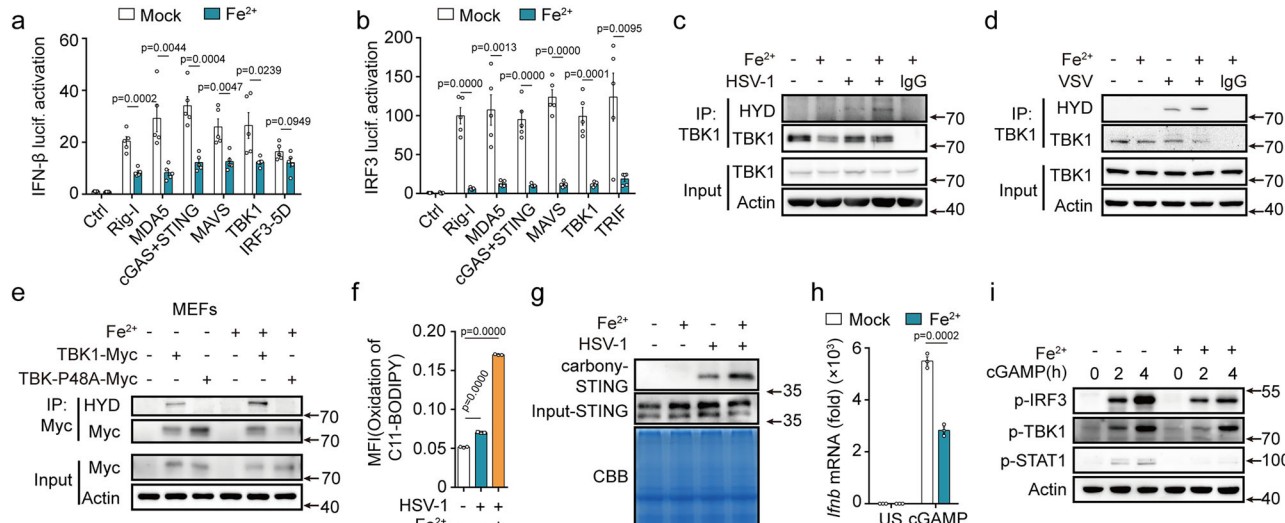

**Fig. 4 | Ferrous iron enhances TBK1 prolyl hydroxylation and STING carbonylation. a, b** Luciferase (lucif.) activity assays of IFN-β reporter activation (**a**) or IRF3 reporter activation (**b**) in HEK293T cells transfected with the indicated adaptors and pretreated with LAL water (Mock) or $Fe^{2+}$ (10 nM) for 1 h. **c, d** Western blot assays of TBK1 prolyl hydroxylation (HYD) in mouse PMs pretreated with $Fe^{2+}$ (10 nM) for 1 h, followed by infection with HSV-1 or VSV, then immunoprecipitation with TBK1 antibody. **e** Immunoprecipitation analysis of lysates from MEFs transfected with Myc-TBK1 (WT), Myc-TBK1-P48A, and pretreated with $Fe^{2+}$ (10 nM) for 1 h, followed by IP with Myc antibody. **f** Flow cytometry analysis of lipid peroxidation in mouse PMs pretreated with $Fe^{2+}$ (10 nM) for 1 h and then infected with HSV-1. Lipid peroxidation is indicated by the MFI (FITC/PE) of lipid peroxidation sensor C11-BODIPY. **g** Western blot assays of carbonylation of STING by selective labeling with m-APA (0.5 mM) in PMs pretreated with LAL water (Mock) or $Fe^{2+}$ (10 nM) for 1 h, followed by HSV-1(MOI:10) infection for 12 h. Coomassie Brilliant Blue (CBB) stain shows the expression of total protein. **h, i** RT-PCR analysis of the *Ifnb* mRNA expression (**h**) or western blot assays of p-TBK1, p-IRF3, and p-STAT1 (**i**) in PMs pretreated with LAL water (Mock) or $Fe^{2+}$ (10 nM) for 1 h and then transfected with cGAMP (5 μg/mL) for 4 h. Data are expressed as the mean ± SEM in **a, b, f,** and **h**. Statistical analyses were performed using two-tailed unpaired Student's t-test. Results were obtained from three independent experiments. US, unstimulated. Unless otherwise specified, cells infection with HSV-1 (MOI:10) for 4 h, VSV (MOI:1) for 8 h. Lysates marker sizes in kDa are indicated on the western blot right. Source data are provided as a Source Data file.

Ferrous iron serves as the raw material for the Fenton reaction to induce lipid peroxidation. Previous studies have reported that lipid peroxidation promotes STING carbonylation and inhibits STING activation[27]. Therefore, we investigated whether ferrous iron targeted STING. Ferrous iron conditions facilitated lipid peroxidation, while DFO alleviated it in HSV-1 infected PMs (Figs. 4f, S5d, e). Therefore, ferrous iron considerably facilitated HSV-1 infection-induced STING carbonylation (Fig. 4g). Similarly, ferrous iron treatment attenuated the endogenous STING ligand cGAMP-induced cytokine expression and the phosphorylation of IRF3, TBK1, and STAT1 (Figs. 4h, i, and S5f). Taken together, these results indicate that ferrous iron targets STING and TBK1 to suppress the host defense against viruses.

## FPN1 enhances host defense and inhibits viral replication

Viruses hijack the membrane iron exporter FPN1 to promote cellular iron accumulation. We examined the potential role of FPN1 in controlling host defense against viruses. FPN1 deficiency attenuated HSV-1, VSV, SeV infection-, and ISD, poly(I:C), cGAMP transfection- induced expression of type I IFNs and RANTES as well as the phosphorylation of IRF3, TBK1, and STAT1 (Figs. 5a–e and S6a–e). Consequently, FPN1 deficiency suppressed ISGs expression and autophagy, resulting in enhanced HSV-1 and VSV replication (Figs. 5f–h and S6f–h). To further confirm the intrinsic role of FPN1, small interfering RNA (siRNA)-silencing experiments were performed in the PMs, and similar results were obtained (Fig. S6i–s). In DFO-treated PMs, FPN1 deficiency had no inhibitory effect on viral infection-induced cytokine expression or the phosphorylation of IRF3 and TBK1 (Figs. 5i, j, and S7a–d). Correspondingly, no differences in HSV-1 and VSV replication were observed in FPN1-deficient PMs after DFO pretreatment (Figs. 5k, l and S7e). These findings indicate that FPN1 facilitates host antiviral responses and limits viral replication by controlling cellular iron homeostasis.

FPN1 facilitated RIG-I, MDA5-, cGAS-STING-, MAVS-, TBK1-, and TRIF-induced signaling, but had no effects on IRF3-induced

IFN-β promoter activation (Fig. 5m). In FPN1-deficient PMs, HSV-1 and VSV infection-induced TBK1 hydroxylation was enhanced (Figs. 5n and S7f). These results indicated that FPN1 targeted TBK1 to promote host defense. The intracellular bacterium *Listeria monocytogenes* also utilizes TBK1 to induce type I IFN responses. We found that ferrous iron treatment and FPN1 deficiency both attenuated *Listeria monocytogenes* infection-induced phosphorylation of IRF3 and TBK1 and the expression of IFN-β and proinflammatory cytokines in macrophages (Fig. S7g–k). However, FPN1 deficiency did not affect the TLR9 pathway (Fig. S7l, m). Collectively, these data indicated that FPN1 selectively promoted TBK1-driven innate immune response.

Next, we investigated the physiological relevance of FPN1 function during viral infection in vivo. VSV infection-induced secretion of IFN-β, TNF-α, and IL-6 were lower in the sera, while VSV replication increased in the spleen and lungs of FPN1-deficient mice (Fig. 6a, b). More severe immune cells infiltration was observed in the lungs of FPN1-deficient mice (Fig. 6c), rendering them more susceptible to viral infections (Fig. 6d). Similarly, in the HSV-1 infected mouse model, FPN1 deficiency suppressed host innate immune responses and facilitated viral replication (Fig. 6e–h). Therefore, FPN1 deficiency attenuated host defense to promote viral replication in vitro and in vivo. These results underscore the indispensable role of FPN1 in governing cellular iron withholding during viral infection, thereby enabling robust host defense against viruses.

## DTX3L attenuates innate responses against RNA virus

Viruses hijack DTX3L to downregulate FPN1 and disrupt iron withholding. Therefore, we examined the role of DTX3L in host antiviral immune response. DTX3L interacts with cGAS[28], and we found that *Dtx3l* deficiency inhibited cGAS pathway (Fig. S8a–c), suggesting that DTX3L may also target cGAS. Therefore, to exclude the effects of DTX3L on host defense caused by other targets, we focused on the

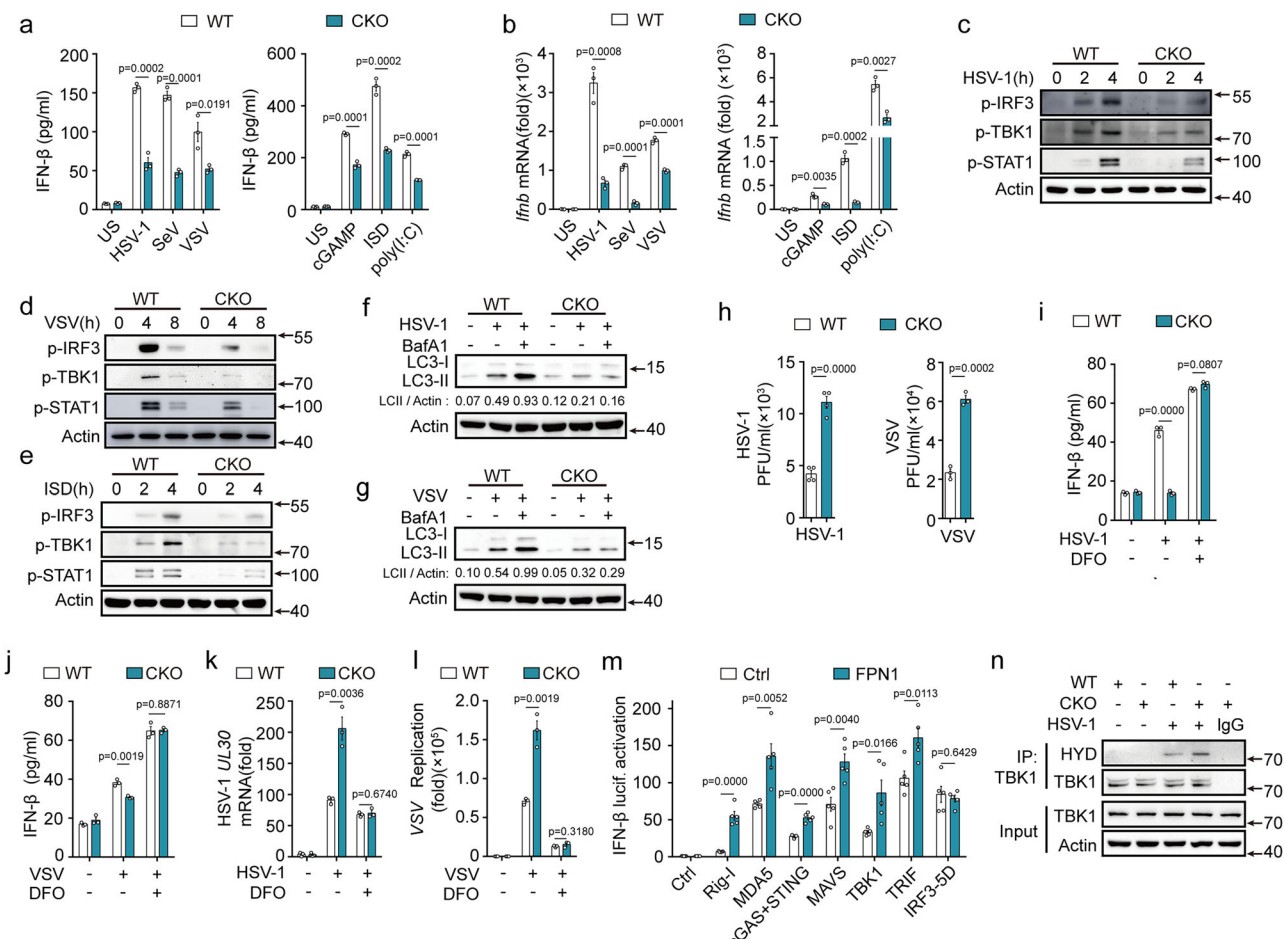

**Fig. 5 | FPN1 enhances type I IFN responses and autophagy. a–e** ELISA analysis (**a**) and RT-PCR analysis (**b**) of IFN-β expression and western blot assays of p-TBK1, p-IRF3, and p-STAT1 (**c–e**) in PMs from WT or *Slc40a1*CKO mice, followed by infection with HSV-1, VSV, SeV, cGAMP (5 μg/mL) transfection for 4 h, ISD (10 μg/mL) transfection for 4 h or poly(I:C) (10 μg/mL) transfection for 8 h. **f, g** Western blot assays of LC3-II and LC3-I in mouse PMs from WT or *Slc40a1*CKO mice, pretreated with BafA1 (100 nM) for 2 h, followed by infection with HSV-1 (**f**) or VSV (**g**). LC3-II/actin level was quantified by measuring band intensities using the "ImageJ" software. **h** Viral titers in mouse PMs from WT or *Slc40a1*CKO mice, followed by infection with HSV-1 or VSV. The viral titers were assessed using the plaque assay. **i, j** ELISA analysis of IFN-β secretion in PMs from WT or *Slc40a1*CKO mice pretreated with DFO (10 μM) for 4 h and then infected with HSV-1 (**i**) or VSV (**j**). **k, l** RT-PCR analysis of the *HSV-1*

*UL30* mRNA (**k**) or the replication of VSV (**l**) in PMs from WT or *Slc40a1*CKO mice pretreated with DFO (10 μM) for 4 h and then infected with HSV-1 for 8 h or VSV for 12 h. **m** Luciferase (lucif.) activity assays of IFN-β reporter activation in HEK293T cells transfected with the indicated adaptors, together with a control vector or Flag-FPN1 plasmid. **n** Western blot assays of TBK1 prolyl hydroxylation (HYD) in PMs from WT or *Slc40a1*CKO mice infected with HSV-1, followed by immunoprecipitation with TBK1 antibody. Data are expressed as the mean ± SEM in **a**, **b**, and **h–m**. Statistical analyses were performed using two-tailed unpaired Student's t-test. Results were obtained from three independent experiments. US, unstimulated. CKO, *Slc40a1*CKO. Unless otherwise specified, cells infection with HSV-1 (MOI:10) for 4 h, VSV (MOI:1) for 8 h, SeV (MOI:1) for 8 h. Lysates marker sizes in kDa are indicated on the western blot right. Source data are provided as a Source Data file.

effects of DTX3L on RLR signaling and innate responses against RNA viruses. As expected, *Dtx3l* deficiency decreased iron accumulation and TBK1 hydroxylation in virus-infected PMs (Figs. 7a and S8d), facilitating viral RNA infection- and poly(I:C) transfection-induced type I IFNs expression and IRF3 phosphorylation (Fig. 7b–f). Similarly, *Dtx3l* knockdown enhanced RLR activation (Fig. S8e–h). Importantly, following ferrous iron administration or FPN1 knockdown, *Dtx3l* deficiency just slightly enhanced VSV infection-induced IFN-β secretion (Fig. 7g, h), suggesting that DTX3L regulated antiviral responses predominantly caused by controlling FPN1 expression and cellular iron homeostasis. In the VSV-infected mouse model, *Dtx3l*-deficient mice exhibited elevated secretion of IFN-β, TNF-α, and IL-6 in sera compared with wild-type mice (Fig. 7i). Consequently, *Dtx3l*-deficient mice displayed decreased viral burden and reduced immune cell infiltration (Fig. 7j–l). By contrast, wild-type mice showed significantly higher susceptibility to VSV-induced lethality than *Dtx3l*-deficient mice (Fig. 7m). These data indicated that DTX3L promotes host defense

against RNA viruses by controlling FPN1-maintained intracellular iron homeostasis.

## Discussion
Perturbation of iron homeostasis is a major strategy in host-pathogen interactions[29]. Iron is required for various vital activities in almost all living organisms, and the competition for iron between the host and pathogens is extremely fierce. Unlike other pathogens such as bacteria and fungi, which have the cellular structure, viruses hijack the host and directly replicate in cells using host materials, including iron[18]. Therefore, intracellular iron is essential for supporting the replication, virulence, and pathogenicity of the invading viruses[18]. Iron plays a double-edged role in cellular biological processes. Various diseases, including infection, cancer, and aging-related diseases, are associated with the disruption of iron homeostasis[30]. In the present study, we showed that viruses increased cellular iron levels to attenuate host defense by targeting TBK1 and STING. Excess cellular ferrous iron

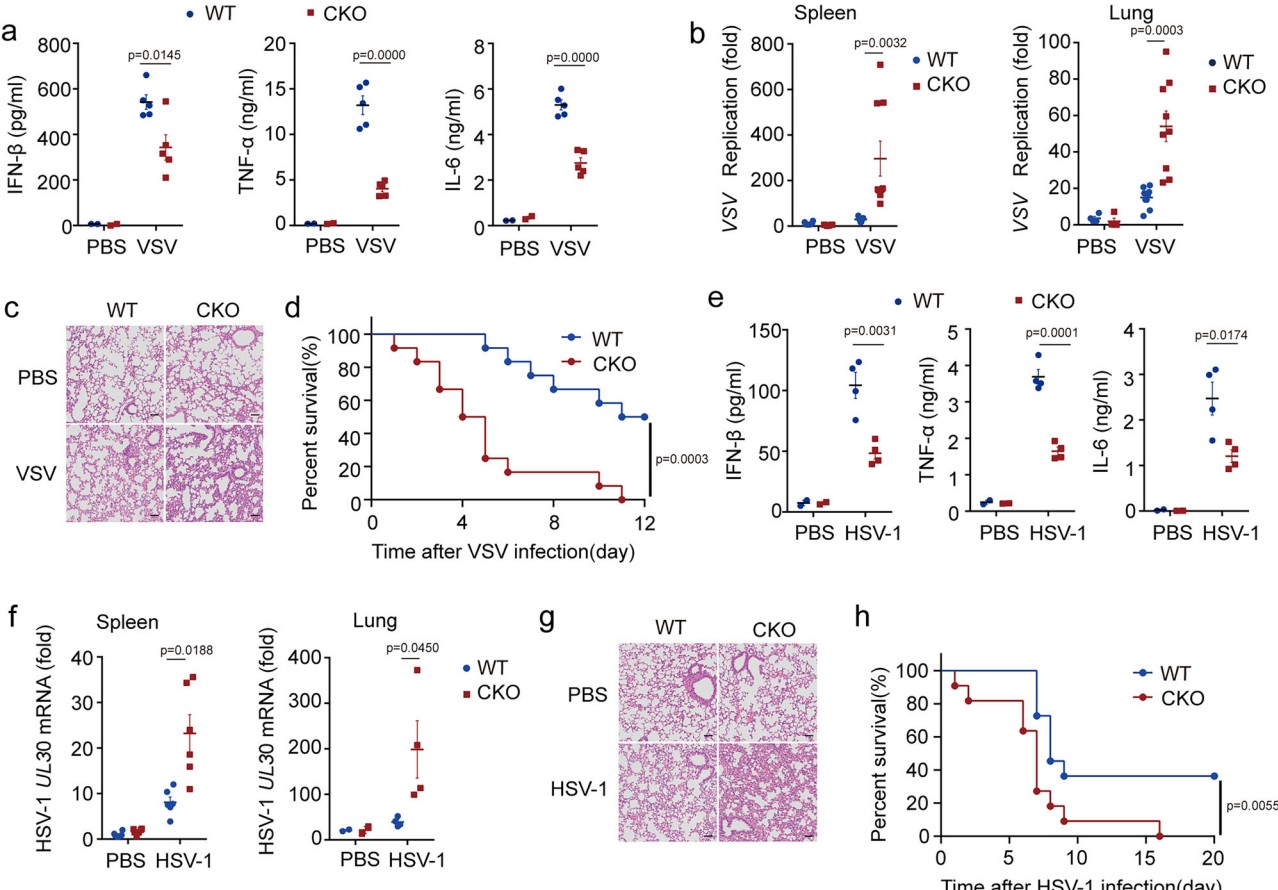

**Fig. 6 | FPN1 inhibits viral replication in vitro and in vivo. a–d** WT or *Slc40a1*[CKO] mice were intraperitoneally injected with VSV (5×10⁷ PFU/mouse) (**a**) ELISA analysis of IFN-β, IL-6, and TNF-α secretion in the serum of mouse infected with VSV for 12 h (*n* = 5 per group). **b** RT-PCR analysis of the replication of VSV in the spleen (left: *n* = 7 per group) and lung (right: *n* = 9 per group) tissues of mouse infected with VSV for 36 h. **c** Hematoxylin and eosin staining of the lung slices of mouse infected with VSV for 36 h. Scale bar, 50 μm. **d** Kaplan–Meier method was used to evaluate survival curves (*n* = 12 per condition), p < 0.001. **e–h** WT or *Slc40a1*[CKO] mice were intraperitoneally injected with HSV-1 (2×10⁷ PFU/mouse). **e** ELISA analysis of IFN-β, IL-6, and TNF-α secretion in the serum of mouse infected with HSV-1 for 8 h. **f** RT-PCR

analysis the *HSV-1 UL30* mRNA in the lung (*n* = 4 per group) and spleen (*n* = 6 per group) tissues of mouse infected with HSV-1 for 24 h. **g** Hematoxylin and eosin staining of the lung slices of mouse infected with HSV-1 for 24 h. Scale bar, 50 μm. **h** Kaplan–Meier method was used to evaluate survival curves (*n* = 11 per condition), *p* < 0.01. Data are expressed as the mean*n* ± SEM in **a**, **b**, **e**, and **f**. Statistical analyses were performed using two-tailed unpaired Student's t-test in **a**, **b**, **e**, and **f** and two-sided log-rank (Mantel–Cox) test in **d** and **h**. Results were obtained from three independent experiments. CKO, *Slc40a1*[CKO]. Source data are provided as a Source Data file.

promotes TBK1 hydroxylation and enhances lipid peroxidation to induce STING carbonylation, resulting in the inhibition of downstream antiviral signaling in macrophages. The activation of TBK1 and STING is critical for the type I IFN response and autophagy, the key host defense mechanisms for viral elimination. Therefore, disrupting host iron homeostasis is beneficial for immune escape and facilitates viral replication in immune cells.

Increasing evidence indicates that intracellular environmental factors control the host's innate responses. For example, the glycolysis intermediate lactate inhibits RLR signaling by directly binding to the MAVS transmembrane domain to prevent MAVS aggregation[31]. Lipid metabolism triggered by LXR agonists induces the expression of the cGAMP-degrading enzyme sphingomyelin phosphodiesterase acid-like 3 A to suppress cGAS[32]. The intracellular polyamine metabolism products spermine and spermidine control B-to-Z DNA transition to orchestrate cGAS activity[33]. Cellular Mn enhances the sensitivity of cGAS to DNA[34], while zinc ions promote cGAS-DNA phase separation[35]. We uncovered the regulatory roles of ferrous iron, another intracellular factor, in cGAS- and RLR-driven antiviral innate immune responses through the suppression of TBK1 and STING.

Iron homeostasis is regulated at multiple levels. At the systemic level, hepcidin serves as the central regulator of iron homeostasis by inhibiting the release of iron from cells. This inhibition primarily targets FPN1 in iron-exporting cells including duodenal enterocytes, recycling macrophages of the spleen and liver, and hepatocytes[18,36]. The HIV-1 protein Nef[37] and the human cytomegalovirus protein US2[38] target HFE, a protein that binds to transferrin receptors and influences hepcidin synthesis. At the cellular level, iron enters the cytoplasm via TfR1 and DMT1, assuming three possible fates: participation in metabolic processes, storage in ferritin, or export through FPN1. As the only known transmembrane export protein, FPN1 is a key determinant of cellular iron levels. Hepcidin binds to FPN1 to mediate its internalization and lysosomal degradation, thereby inducing cellular ferrous iron accumulation[19,20]. The E3 ubiquitin ligase RNF217 promotes FPN1 degradation and impairs iron homeostasis[21]. Recent studies have reported that the hepatitis C virus (HCV) NS3-4A serine protease mediates FPN1 cleavage, impairing iron efflux and causing iron accumulation in HCV-infected liver cells[39]. However, the precise mechanism by which viruses hijack host cells to affect FPN1 expression remains unclear. By screening for

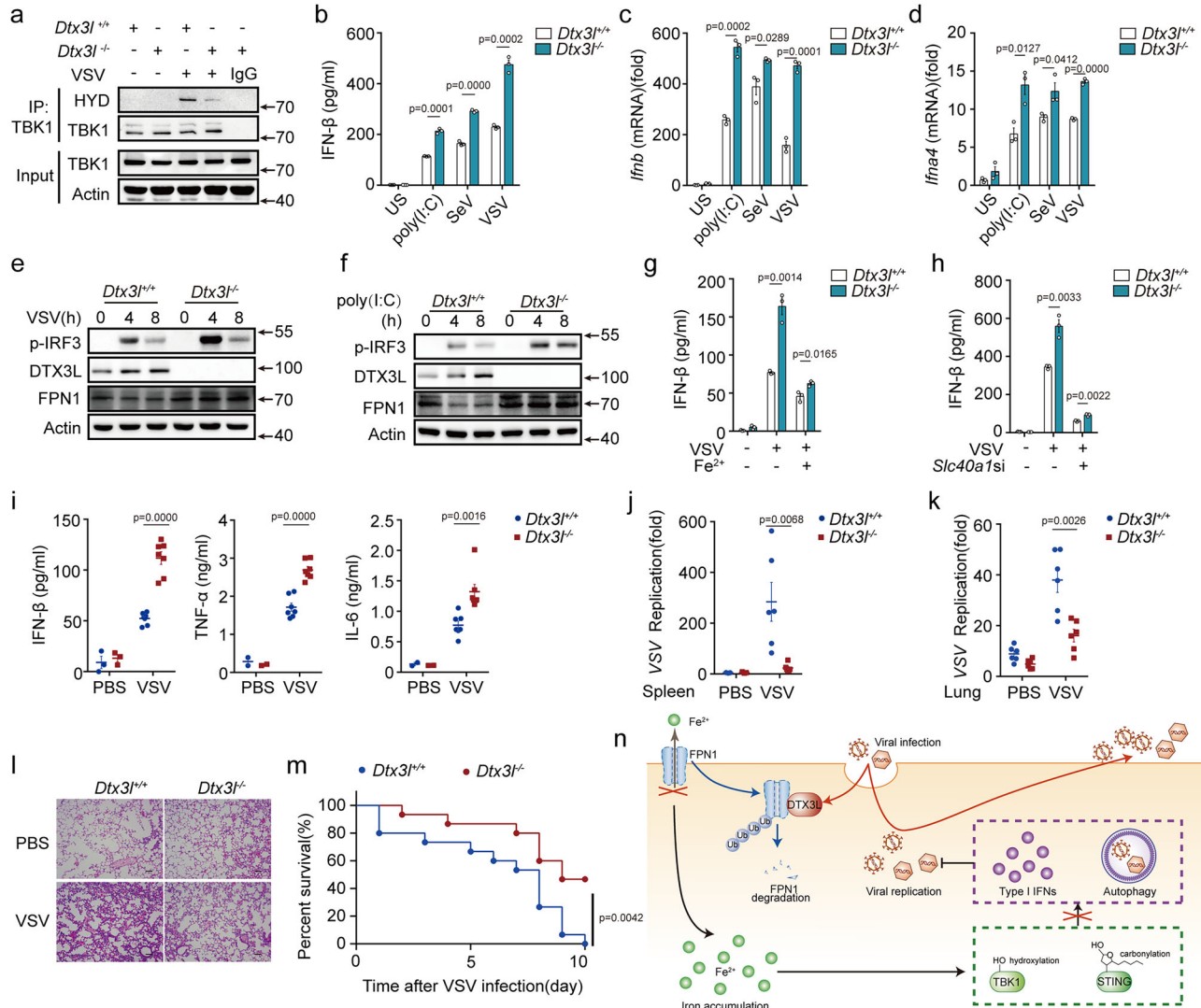

**Fig. 7 | DTX3L attenuates innate responses against RNA viruses. a** Western blot of TBK1 prolyl hydroxylation (HYD) in *Dtx3l*[+/+] or *Dtx3l*[−/−] mouse PMs, infected with VSV (MOI: 1) for 8 h, followed by immunoprecipitation with TBK1 antibody. **b−f** ELISA analysis (**b**), RT-PCR analysis (**c, d**) of IFN-β expression (**b, c**) and *Ifna4* expression (**d**), or western blot of p-TBK1, p-IRF3, and p-STAT1 (**e, f**) in PMs from *Dtx3l*[+/+] or *Dtx3l*[−/−] mice, followed by infected with VSV, SeV, or poly(I:C) (10 μg/mL) transfection for 8 h. **g, h** ELISA assay of IFN-β secretion in PMs, pretreated with LAL water (Mock) and Fe[2+] (10 nM) for 1 h (**g**), or *Ctrl* siRNA and *Slc40a1* siRNA-transfected PMs (**h**) from *Dtx3l*[+/+] or *Dtx3l*[−/−] mice, followed by infection with VSV. **i−m** *Dtx3l*[+/+] or *Dtx3l*[−/−] mice were intraperitoneally injected with VSV (5×10[8] PFU/mouse). **i** ELISA analysis of IFN-β, IL-6, and TNF-α secretion in the serum of mouse infected with VSV for 12 h (n = 7 per group). **j, k** RT-PCR analysis of the replication of VSV in the lung (n = 6 per group) and spleen (n = 6 per group) tissues of mouse

infected with VSV for 36 h. **l** Hematoxylin and eosin staining of the lung slices of mouse infected with VSV for 36 h. Scale bar, 50 μm. **m** Kaplan−Meier method was used to evaluate survival curves (n = 15 per condition), p < 0.01. **n** The iron withholding mechanism of FPN1 inhibits viral replication. During viral infection, FPN1 interacts with the E3 ubiquitination ligase DTX3L and is degraded via the proteasomal pathway. FPN1 promotes viral replication through the process of autophagy and enhances type I IFN signaling by targeting TBK1 hydroxylation. Data are expressed as the mean ± SEM in **b−d, g−k**. Statistical analyses were performed using two-tailed unpaired Student's t-test in **b−d, g−k** and two-sided log-rank (Mantel−Cox) test in m. Results were obtained from three independent experiments. US, unstimulated. Unless otherwise specified, cells infection with VSV (MOI:1) for 8 h, SeV (MOI:1) for 8 h. Lysates marker sizes in kDa are indicated on the western blot right. Source data are provided as a Source Data file.

upregulated genes in virus-infected macrophages, we identified the virus-induced host E3 ubiquitin ligase DTX3L as an inhibitor of FPN1 expression. As an important E3 ligase in DNA damage response[40], DTX3L have been recently implicated in enhancing interferon signaling by mediating histone H2BJ monoubiquitination[41] and TBK1 K63-linked ubiquitination[42]. In our study, DTX3L selectively facilitates the K48-linked ubiquitination and protein degradation of FPN1, thus disrupting host iron withholding during viral infection. These different results indicated that different types of ubiquitination modifications, different cell types and viruses led to different functions of DTX3L. Compared with the host systemic iron regulator

hepcidin secreted by the liver, which degrades FPN1 within 4 h following infection[19,43], DTX3L is rapidly induced after viral infection, robustly promoting FPN1 degradation. Consequently, early-stage ferrous iron accumulation in macrophages is beneficial for quick and efficient viral replication.

In summary, our study uncovered a new mechanism by which viruses manipulate cellular iron levels to suppress host antiviral defenses, establishing a link between iron homeostasis and innate immune responses. Viruses disrupt iron withholding by hijacking FPN1, thereby obtaining more iron to meet the needs of replication and suppressing the activation of TBK1 and STING. Our results indicate that

FPN1-maintained intracellular iron homeostasis is required for the manipulation of cGAS- and RLR-dependent downstream innate responses, revealing an additional immune escape mechanism and thereby providing promising therapeutic targets for the treatment of viral diseases.

## Methods

### Ethics statement

All animal experiments were performed in compliance with the National Institutes of Health Guide for the Care and Use of Laboratory Animals with approval from the Scientific Investigation Board of the School of Basic Medical Science, Shandong University, Jinan, Shandong Province, China. The approval number is ECSBMSSDU2021-2-048.

### Mice

*Slc40a1*<sup>flox/flox</sup> (stock number: 53945), *Lyz2*<sup>cre</sup> (stock number: 004781), and *Dtx3l*<sup>−/−</sup> mice (stock number: 209200) on C57BL/6 background were obtained from Cyagen Biosciences Inc. (Guangzhou, China). *Ifnar1*<sup>−/−</sup> mice (JAX stock #028288) were obtained from the Jackson Laboratory. *Slc40a1*<sup>flox/flox</sup> mice were crossed with *Lyz2*<sup>cre</sup> transgenic mice to generate "*Slc40a1* <sup>CKO</sup>" mice with FPN1 deficiency in myeloid cells. C57BL/6 mice were purchased from the Vital River Laboratory Animal Technology Co. (Beijing, China). All animals were kept at specific pathogen-free (SPF) levels with 40%–70% humidity and exposed to 12 h light (23 °C) and 12 h dark (21 °C) cycles daily. All animals were 6–8 weeks old, and both male and female mice were used.

### Cells

To obtain mouse primary PMs, male and female C57BL/6 mice (6–8 weeks old) were intraperitoneally injected with 3% Brewer's thioglycolate broth (3 g of starch, soybean grain size beef paste, dissolved in 100 mL distilled water, autoclaved and stored at 4 °C, used after overnight storage). Three days later, peritoneal exudate cells were harvested and incubated. 4 h later, non-adherent cells were removed, and the adherent monolayer cells were used as peritoneal macrophages (PMs). Human THP-1 (Cat#HIB-202), human embryonic kidney (HEK293T) cells (Cat#CRL-3216) and Vero cells (Cat#CRL-1586) were obtained from the American Type Culture Collection (Manassas, VA, USA). Phorbol myristate acetate (PMA, 100 ng/ml) -activated THP-1 cells were used as human macrophages. Mouse embryonic fibroblasts (MEFs) were generated from pregnant female mice at 13–14 days of gestation. Briefly, embryos were removed from the uterus of timed pregnant females and dissected from the decidua and parietal yolk sac using fine forceps, leaving the ectoplacental cone attached to the egg cylinder, and the fetal viscera, head, and limbs were excised. The rest of the embryonic tissue was minced and incubated with 0.25% trypsin-ethylenediaminetetraacetic acid solution for 30 min at 37 °C. MEFs were cultured and expanded for subsequent experiments. All the cells were cultured at 37 °C under 5% $CO_2$. PMs, MEFs, Vero cells and HEK293T cells were cultured in Dulbecco's modified Eagle's medium (DMEM) supplemented with 10% fetal bovine serum (FBS). THP-1 cells were cultured in Roswell Park Memorial Institute medium (RPMI) supplemented with 10% FBS.

For mouse spleen single-cell suspension, spleens were harvested and dissociated through 70-µm cell strainers. Cells were collected and centrifuged at $300 \times g$ for 5 min. The supernatant was discarded, and the cell pellet was subjected to red blood cell lysis using 2–3 mL of red blood cell lysis buffer (CWBIO, CW0613). Cells were washed 3 times with PBS, filtered again with 70-µm strainers and counted. The cell concentration was adjusted to $1 \times 10^6$ cells/mL. Prepared for staining in PBS with 2% FBS at 4 °C.

For mouse single-cell suspension from liver, livers were removed and minced into small pieces, which were shaken in the digestion buffer which contained 0.05% collagenase type I diluted by TESCA buffer (50 mM TES, 0.36 mM $CaCl_2$, pH 7.4) and 0.02% DNase I in Hank's Balanced salt solution (HBSS) at 37 °C for 20 min, homogenized and filtered through a 70-µm cell strainer. For elimination of hepatocytes, the cell suspension was centrifuged at 50 g for 5 min at room temperature, and then supernatant was collected, washed in PBS and resuspended in a 40% Percoll gradient (P4937; Sigma-Aldrich, Saint Louis, MO). The cell suspension was gently overlaid onto 70% Percoll and centrifuged at 650 g for 30 min. liver single-cell suspension were collected from the interface. The cells were resuspended, filtered again with a 70-µm strainer and counted. The cell concentration was adjusted to $1 \times 10^6$ cells/mL. Prepare for staining at 4°C in PBS with 2% FBS.

### Reagents and antibodies

Ferrous chloride, 3-ethynylaniline (m-APA; 498289), PMA (79346), poly (I: C), Z-Leu-Leu-Leu-al (MG132, C2211) and BioTracker575 Red Fe²⁺ (SCT030) were obtained from Sigma-Aldrich (St. Louis, MO, USA). Deferoxamine mesylate (S5742), QNZ (S4902), 3-methyladenine (3-MA: S2767), and chloroquine (S6999) were purchased from Selleck Chemicals. JSH-23 (S7351) was purchased from Selleck Chemicals (Houston, TX). Itacitinib (T3998) was purchased from Targetmol. Iron (II) sulfate heptahydrate (F7002), biotin-alkyne (B171422), and biotin-PEG2-azide (B122224) were obtained from Aladdin. Meanwhile, 2′3′-cGAMP (tlrl-nacga23-1) and ISD (tlrl-isdn) were obtained from InvivoGen. Cycloheximide (CHX: A8244) and Cell Counting Kit-8 (CCK-8) (cat. #K1018) were obtained from APExBIO Technology and used to quantitatively measure cell viability. Chromatin immunoprecipitation grade protein G magnetic beads (cat. #9006S) were purchased from Cell Signaling Technology (Danvers, MA, USA). Anti-Flag (cat. #F1804) and anti-HA (cat. #H3663) were purchased from Sigma-Aldrich (St. Louis, MO, USA). Anti-β-actin (cat. #sc-8432), CD11b-CY7 (cat. #101215), F4/80-PE (cat. #123110), and Ly6C-APC (cat. #128015) were purchased from BioLegend (San Diego, CA, USA). Anti-p-TBK1 (5483), anti-p-IRF3 (Ser396, 4947), anti-p-STAT1 (9167), anti-mouse IgG (7076), anti-rabbit IgG (7074), and streptavidin (magnetic bead conjugate) (5947) antibodies were purchased from Cell Signaling Technology. Anti-DTX3L (ab70621), anti-Ferritin (ab75973), and anti-TFR1 (ab214039) were purchased from Abcam. Alexa Fluor 633 (A-21071) and Alexa Fluor 488 (A-11059) were purchased from Thermo Fisher Scientific and used for immunoprecipitation, while DMT1 (sc-166884), TFR1 (sc-271485), TFR2 (sc-32271) (for western blot), and horseradish peroxidase-conjugated secondary antibodies were purchased from Santa Cruz Biotechnology. Anti-TFR2 (A9845) (for flow cytometry) was purchased from ABclonal Technology (Wuhan, China). Anti-LC3 (14600-1-AP), anti-FPN1(26601-1-AP) (for western blot) were purchased from Proteintech (Wuhan, China). Anti-FPN1 (NBP1-21502) (for flow cytometry) was purchased from Novus Biological (St. Louis, MO, USA). Anti-FPN1 (MTP11-A) (for immunoprecipitation) was purchased from Alpha Diagnostic Intl. (San Antonio, TX, USA). TNT Quick Coupled (Transcription/Translation System, L1170) was purchased from ProMega (Madison, WI, USA).

### Viruses

SeV was purchased from the China Center for Type Culture Collection (China). The VSV-GFP and HSV-1 were obtained from X. Cao (Second Military Medical University, Shanghai, China).

For propagation of VSV and HSV-1, Vero cells were infected with virus at a MOI of 2, and the viruses produced by one replication cycle were harvested. For purification of virus, the cell cultures were first freeze-thawed twice, and the supernatants were clarified by centrifugation at $3000 \times g$ for 1 h, followed by pelleting of the virus by

ultracentrifugation at 45,000 g for 1 h. HSV-1 and VSV were propagated and titrated by the plaque assay on Vero cells.

## Plaque assay

Supernatants from HSV-1 and VSV infected cells were diluted and added to overgrown Vero cells cultured in 24-well plates. 1 h later, the supernatants were removed, and the cells were washed twice with phosphate-buffered saline (PBS, Procell Life Science & Technology). The cells were incubated in DMEM containing 1% methylcellulose(CAS: #9004-67-5, Sigma-Aldrich) for 48 h (VSV) or 72 h (HSV-1), fixed with 4% paraformaldehyde (Servicebio, Wuhan, China, G1101) for 15 min, and stained with crystal violet (Beyotime Biotechnology, C0121) for 30 min. Plaques were counted, and viral titers were determined as pfu/mL.

## Plasmids and transfection

The Myc-FPN1, Flag-FPN1, Flag-TRIM21, Flag-MARCH5, Flag-DTX3L, and Flag-RNF114 plasmids were purchased from Vigene Biosciences. The Flag-DTX3L (C651A and C564A) and Myc-TBK1 (P48A) mutants were generated from Flag-DTX3L and Myc-TBK1 plasmids using the KOD-Plus-Mutagenesis kit (Toyobo, Osaka, Japan, cat. #SMK-101). All constructs were confirmed by DNA sequencing. Plasmids (2–4 µg plasmids in 6-wells plates, 4–6 µg in 6 cm dishes) were transiently transfected into HEK293T cells and MEFs using the Lipofectamine 2000 reagent (Invitrogen) (w/v = 1:1) according to the manufacturer's instructions.

SiRNA (2–4 µg siRNA in 24-wells plates, 2–4 µg in 12-wells plates, 4–6 µg in 6-wells plates) duplexes were transfected into mouse PMs using INTERFERin reagent (PolyPlus, cat. #409-10) (w/v = 1:2), according to the manufacturer's instructions.

IFN-β reporter plasmid and expression plasmids for RIG-I, MDA5, MAVS, TRIF, TBK1, IRF3 5D, and HA-Ub were described previously[33]. For luciferase activity assays, cells are cultured in 96-well plates, and plasmids (up to 350 ng) transfected by JetPEI (PolyPlus, cat. #10110) (w/v = 1:2), according to the manufacturer's instructions.

ISD (10 µg/mL), poly(I:C) (10 µg/mL), and cGAMP (5ug/mL) were transfected into mouse PMs using Lipofectamine 2000 (w/v = 1:1), according to the manufacturer's instructions.

## Viral infection in vivo

A total of 8-week-old C57BL/6 J mice were intraperitoneally injected with HSV-1 ($2 \times 10^7$ PFU/mouse) or VSV ($5 \times 10^7$ or $5 \times 10^8$ PFU/mouse). For wild-type (WT) and *Slc40a1*[CKO] mice, 6-8-week-old and sex-matched littermates were infected with HSV-1 or VSV. Mouse serum was collected and subjected to enzyme-linked immunosorbent assay (ELISA) 8 h after viral infection. Mice were euthanized 36 h after HSV-1 or 48 h after VSV infection, and the spleen and lung tissues were harvested for quantitative real-time PCR (RT-PCR). Lungs from control and virus-infected mice were dissected, fixed in 10% phosphate-buffered formalin, embedded in paraffin, sectioned, stained with hematoxylin and eosin, and examined under a light microscope to assess for histological changes. The mice's survival was monitored following viral infection.

## ELISA and real-time-PCR (RT-PCR)

ELISA kits for IFN-β (BioLegend, San Diego, CA), TNF-α (Dakewe Biotech, Shenzhen, China), and IL-6 (Dakewe Biotech) were used to quantify the respective cytokines and compounds in sera of mice or supernatants of mouse PMs ($5 \times 10^5$ cells /well in 24-well plates). Malondiadehyde (MDA) ELISA kit (#AB287797, Abcam, Cambridge, MA) were used to quantify the MDA production in supernatants of mouse PMs ($2 \times 10^6$ cells /well in 6-well plates).

RNA from mouse PMs ($1 \times 10^6$ cells /well in 12-well plates) was extracted using the RNA fast200 Extraction kit (Fastagen, Shanghai, China) according to the manufacturer's instructions. For mouse tissues, total RNA was extracted using TRIzol reagent (Invitrogen)

according to the manufacturer's instructions. RNA was reverse transcribed using reverse transcriptase (Vazyme, Nanjing, China). RT-PCR analysis was performed using the StepOnePlus Real-Time PCR System and SYBR RT-PCR kits (Roche, Switzerland). The primers used for qPCR were presented in Table S1. The data were standardized based on β-actin expression.

## Western blot

The cells ($2 \times 10^6$ cells/well in 6-well plates) were lysed using radio-immunoprecipitation assay buffer (Pierce, Thermo Fisher Scientific Inc.) supplemented with a protease inhibitor cocktail (Sigma). The protein concentrations were measured using the Pierce BCA Protein Assay Kit (Thermo Fisher Scientific). Equal amounts of the extracts were separated by sodium dodecyl sulfate polyacrylamide gel electrophoresis (SDS-PAGE) and transferred in polyvinylidene difluoride membranes (Millipore, Burlington, MA, USA) for immunoblot analysis. The values quantitated by measuring band intensities using the "ImageJ" software were normalized to those of actin.

## Immunoprecipitation (IP)

Cells ($3-5 \times 10^6$ cells/well in 6 cm dishes) were lysed with IP lysis buffer, which contained 1.0% (v:v) NP-40 (Solarbio, N8030), 50 mM Tris HCl pH 7.4 (Solarbio, Z9912), 50 mM EDTA (Solarbio, E1170), 150 mM NaCl, and a protease inhibitor cocktail. For all the IP detecting ubiquitination, cells were pretreated with MG132 (10 µM) for 4 h before collected. After 15 min centrifugation at $12,000 \times g$ at 4 °C, protein concentrations in the lysates were measured with a Pierce BCA Protein Assay Kit (Pierce, Rockford, IL), and then supernatants were collected and incubated with 40 µL protein G Plus-Agarose IP reagent together with 1 µL specific antibody overnight at 4 °C. Beads were washed five times with IP lysis buffer. Immunoprecipitates were eluted by boiling with 1% (w/v) SDS sample buffer.

For all the IP assay associated with FPN1, cells supernatants were collected and incubated with 1 µL specific antibody together with 40 µL magnetic beads (Cell Signaling Technology) overnight at 4 °C. Immunoprecipitates were eluted by Acidic elution (0.1 M Glycine-HCl, pH3.0) and immediately add 10 µL of neutralization solution (0.5 M Tris-HCl, pH 7.4, 1.5 M NaCl) without boiling.

## Flow cytometry analyses

The collected single cell suspensions were washed 3 times with PBS containing 2% FBS and stained by surface antibodies (DMT1(1:50), TFR1(1:100), TFR2(1:100), FPN1(1:200)) for 1 h at 4 °C, then cells washed 3 times with PBS containing 2% FBS, followed by incubation with anti-AF488 (1:50, A11008(R), A11059(M), ThermoFisher Scientific), or AF633 (1:50, A-21071(R) ThermoFisher Scientific), secondary antibody at room temperature for 30 min, then prepared the cells at 4 °C in PBS with 2% FBS. Anti-mouse CD11b-CY7 (BioLegend, 1:100), anti-mouse F4/80-PE (BioLegend, 1:200), and anti-mouse Ly6C-APC (BioLegend, 1:100) were used to detect F4/80+ splenic macrophages, anti-mouse F4/80-PE (BioLegend, 1:100) were used to detect F4/80+ liver macrophages. Intracellular staining was performed with specific probes (BioTracker575 Red Fe[2+], Calcein, C11-BODIPY) according to the manufacturer's instructions (see below). Samples were assessed using CytExpert (Beckman Coulter, Inc) in flow cytometry tubes to detect the median fluorescence intensity (MFI) of different target molecules and analyzed using FlowJo v10.8.1 (FlowJo). The representative flow cytometry gating strategy images are described in Fig. S9.

## Quenchable iron pool (QIP) quantification Assays

The QIP assay was performed as previously described[15]. In detail, PMs ($2 \times 10^6$ cells/well in 6-well plates) were washed 3 times with PBS and stained with 300 µL of Calcein-AM (BioLegend, 1 µM) in PBS for 15 min at 37 °C. After two PBS washing steps, PMs were

treated with 300 μL PBS or 300 μL FeHQ solution (5 μM $FeCl_2$ and 10 μM 8-hydroxyquinoline in PBS, all from Sigma-Aldrich) for 30 min at 37 °C. The Calcein median fluorescence intensity (MFI) of the PMs was detected on a CytoFLEX (Beckman) and analyzed using FlowJo software version 10 (FlowJo). The QIP of PMs per experimental condition was calculated as the difference between the MFI (PBS treated) and MFI (FeHQ treated). At least three technical replicates were analyzed under the PBS-treated and FeHQ-treated conditions.

To quantify QIP in spleen or liver macrophages in vivo, prepared mouse spleen or liver single-cell suspension as previously shown, resuspended in 1 mL DMEM/10% FBS, and counted. The cell concentration was adjusted to $1×10^6$ cells/mL. The cells were transferred into 1.5 mL reaction tubes and harvested at 1000 g for 5 min at 4 °C. After removing the supernatant, 50 μL of FACS buffer containing CD11b-Cy7, F4/80-PE, and Ly6C-APC (BioLegend) were added, and splenocytes were stained for 30 min on ice and harvested at 1000 g for 5 min at 4 °C. For liver tissues, only anti-F4/80-PE was used. The supernatant was removed, and single-cell suspension were stained in 200 μL of 0.1 μM Calcein-AM (BioLegend) for 15 min at 37 °C (mixed 5 times). After washing with PBS twice, the single-cell suspension were either left untreated in 200 μL of PBS or were treated with 200 μL of FeHQ solution for 30 min at 37 °C (mixed 5 times). After cells harvested at 1000 g for 5 min at 4 °C, the supernatant was removed, and single-cell suspension were resuspended in 300 μL of FACS buffer. Splenocyte QIP quantification was performed using the identical procedure outlined for PMs. The representative flow cytometry gating strategy images are described in Fig. S9.

### Iron content analysis

Iron detection through Iron Assay Kit (Abcam, Cambridge, MA), $4\text{-}6 × 10^7$ cells in 2–3 of 15 cm culture dishes were washed twice with PBS, extracted with 1 mL assay buffer, and centrifuged at $16,000 × g$ for 10 min; then, the supernatant was collected. Fe reducer was able to reduce $Fe^{3+}$ to $Fe^{2+}$. So, for $Fe^{2+}$ assay, added assay buffer only. For total iron ($Fe^{2+}$ and $Fe^{3+}$), added iron reducer. The level of $Fe^{3+}$ was calculated by subtracting $Fe^{2+}$ iron from total iron. After 30 min incubation at room temperature, iron reaction solution was added to the samples, and further incubation was carried out for another 60 min in the dark at room temperature. The absorbance was measured at an optical density of 593 nm using a Microplate Reader FlexA-200. Intracellular iron content was calculated using a standard curve and normalized to the total number of cells.

$Fe^{2+}$ detection through flow cytometry, PMs ($4\text{-}6 × 10^7$ cells in 6-well plates) were washed twice with HBSS (MacGene Technology #CC016), then added 5 μM BioTracker 575 Red staining stock solution (1:200, diluted by DMSO) and incubated for 1 h. The cells were suspended in HBSS and the MFI of cells were detected by using CytExpert (Beckman Coulter, Inc).

### RNA-Seq experiment and analysis

After HSV-1, VSV, or SeV infection, total RNA was extracted from PMs using the TRIzol reagent (Invitrogen, 15596018). The extracted RNA was sent to Capital Bio Technology for assays and deep sequencing. The extracted RNA was used to prepare a cDNA library according to standard Illumina RNA-seq instructions. The generated cDNA library was sequenced in a $2 × 10^5$ bp paired-end layout using an Illumina HiSeq2500. Library preparation and sequencing were performed in Novogene (Beijing, China).

To determine genes expression changes after virus infection (HSV-1, VSV, and SeV) by RNA-Seq experiment. First, the RNA-seq data were processed by the edgeR package to analyze the paired differential genes, the p values underwent FDR correction. Among 291 genes with E3 activity, which were chosen as significantly upregulated after viral infection, with $\log_2FC$ of >2 and FDR of <0.001 as statistical boundaries. Finally, the significantly upregulated E3 activity genes were visualized using a schematic diagram.

### Determination of lipid peroxidation by C11-BODIPY (FITC/PE)

Following the indicated pretreatment of PMs ($2 × 10^6$ cells/well in 6-well plates), 50 μM of C11-BODIPY (Thermo Fisher Scientific, #D3861) was added and incubated for 30 min. The cells were washed to 3 times with PBS to remove excess C11-BODIPY. PE (non-oxidized state) and FITC (oxidized state) were detected because oxidation of the polyunsaturated butadiene group portion of C11-BODIPY could shift the fluorescence emission peak from PE to FITC. Then, MFI was analyzed by flow cytometry.

### Determination of STING carbonylation

To ascertain STING carbonylation, $2 × 10^6$ cells/well in 6-well plates were lysed in 0.1% Triton X-100/PBS and centrifuged at 15,000 g for 30 min to remove the cell debris (sonicated at 4°C). The proteome was maintained at 2 mg/mL and treated with 0.5 mM m-APA at pH 5.0 for 1 h. The protein was precipitated with methanol and chloroform and resuspended in 200 μL 0.4% SDS/PBS. The proteome was subjected to a reaction with 1 mM of $CuSO_4$, 100 μM of TBTA ligands (TentaGel TBTA (Sigma Aldrich, #696773)), 100 μM of biotin-alkyne (Aladdin, #B171422), and 1 mM TCEP (Aladdin #T724721) for 1 h at 25 °C. After the second precipitation, the target protein was enriched with streptomycin-conjugated agarose beads for 3 h, washed with PBS 3 times, and then separated from the target protein by loading buffer (0.1% SDS IP lysis buffer) at 95 °C for 20 min. Western blot was performed for detection.

### Immunofluorescence staining and confocal analysis

Macrophages ($2\text{-}5 × 10^5$ cells/well in 24-well plates) were infected with HSV-1 or VSV. FPN1 (1:200) and DTX3L (1:200) were stained with a secondary antibody conjugated to either Alexa Fluor 633 (1:50) or Alexa Fluor 488 (1:50), and the nuclei were stained with DAPI (Servicebio #G1012). The cells were examined under a confocal laser microscope (LSM980; Carl Zeiss).

### Statistical analysis

GraphPad Prism software was used for statistical analysis. All data were expressed as the mean ± SEM of two or three experiments and were subjected to unpaired two-tailed Student's t-test. A $p$-value of <0.05 was considered significant.

### Reporting summary

Further information on research design is available in the Nature Portfolio Reporting Summary linked to this article.

## Data availability

The RNA-seq datasets generated during the study are available from the Gene Expression Omnibus (GEO) Database (http://www.ncbi.nlm.nih.gov/geo/) under super series number GSE280144. The main data supporting the results in this study are available within the article and its Supplementary Information. The raw and analysed datasets generated during the study are available for research purposes from the corresponding authors on reasonable request. Source data are provided with this paper.

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

## Acknowledgements

We thank the Translational Medicine Core Facility of Shandong University for the consultation and instrument availability that supported this work. This work was supported by grants from the National Natural Science Foundation of China (grant no. 82125020 to W.Z., 82321002 to C.G. and 32470925 to C.Z.), and Shandong Provincial Nature Science Foundation (ZR2023ZD57 to W.Z.).

## Author contributions

W.Z. and C.Z. designed the experiments and wrote the manuscript. L.T., J.W., Y.M., and C.W. performed the majority of the experiments, while Q.L. and Y.F. analyzed the data. C.Z., Y.Q., and H.S. assisted contributed to the experiment process and provided technical assistance. C.G. provided valuable expertise and advice. W.Z. conceived the project and provided overall directions.

## Competing interests

The authors declare no competing interests.
