## [Peer Review file · Nature Communications]

Viruses hijack FPN1 to disrupt iron withholding and suppress host defense

Corresponding Author: Professor Wei Zhao

Version 0:

Reviewer comments:

Reviewer #1

(Remarks to the Author)

Zhao and colleagues describe a novel mechanism of viral escape to immune response at the cellular levels by modulating iron levels. Viral infection leads to the upregulation of DTX3L that in turn, induced the degradation of ferroportin, causing intracellular iron accumulation. Excessive Fe²⁺ levels reduced type I IFN responses. The manuscript is clearly organized and well written. Nevertheless, I have two major concerns:

- 1) Intracellular iron is mostly in the Fe³⁺ form (bound to ferritin) as the labile iron pool (Fe²⁺) must be kept minimal to avoid cellular damage. The study addressed the effects of Fe²⁺ ions but did not study the effect of infection on ferritin levels and Fe³⁺ levels.
- 2) The authors resorted to in vivo models of viral infection to confirm their in vitro findings. However, when looking to iron metabolism/ homeostasis at the systemic level, hepcidin-mediated effects should be considered. Hepcidin is upregulated during inflammatory conditions, such as viral infections, and binds to FPN1 leading to its degradation. Thus, hepcidin levels and its effects on FPN1 levels should be determined to dissect the relevance of the DTX3L-mediated FPN1 degradation in the in vivo models of viral infection.

Minor concerns:

1. Line 93 "These findings indicate that viruses disrupt host iron withholding and cause iron accumulation in the cells." There is no evidence in the results to support the conclusion that viruses disrupt host iron withholding at this point.
2. Line 97 "a significant downregulation of the sole cellular iron exporter, FPN 1". Have the authors performed a quantitative analysis followed by statistical testing to support this statement? Along this line, the amount of FPN1 protein at 12h post-IFN β stimulation (Fig 1i) is similar to that observed for the same timepoint but for viral infections (Fig 1e, 1f), and the authors consider that IFN β has not direct effect on FPN1 levels. Moreover, the kinetics of FPN1 downregulation seem to differ between a primary cell (mouse PM), and a cell line (THP-1). I would like the authors to give a rationale for the difference.
3. Lines 111-113 "Notably, this effect was reversed by the proteasome inhibitor MG132, but not by the lysosome inhibitors chloroquine and 3methyladenine(3-MA) (Fig.2c)." The data do not depict what is described in those lines, namely regarding chloroquine effect on FPN1 levels.
4. Lines 130-134: Does the treatment with NF- κ B inhibitors (JSH-23 and QNZ) and JAK1 inhibitor itacitinib affect FPN1 levels?

Reviewer #2

(Remarks to the Author)

Cells maintain iron homeostasis for normal cellular physiology and functions. Viruses are known to hijack iron metabolism to facilitate their replication. In this manuscript, Li Tong and colleagues report a negative role of iron in antiviral innate immunity. The authors show that either DNA or RNA virus infections induce polyubiquitination and degradation of ferroportin (FPN1), leading to intracellular iron accumulation rapidly. FPN1 ubiquitination and degradation is mediated by a host E3 ligase, DTX3L, which expression is upregulated following viral infection. DTX3L interacts with FPN1 and mediates its K48-linked polyubiquitination. Excessive iron suppresses type I IFN responses and autophagy by promoting TBK1 hydroxylation and STING carbonylation. Therefore, FPN1 deficiency results in impaired innate immunity while enhanced viral replication in vitro and in vivo. In contrast, DTX3L deficiency has the opposite effect. The study is comprehensive; the results are interesting and novel.

Specific comments:

1. In all the viral infection/ligand stimulation experiments/figures, the MOI and time of infection/stimulation should be specified in their corresponding figure legends (including supplemental figures). "US" should be spelled out in each figure legend.
2. Fig.1g,h, l: the time of infection/ligand stimulation should be indicated. There are two DMT1 bands in Fig. 1e, f, but one band in Fig. 1i and Fig. S1i and j. Please clarify this discrepancy and indicate the correct band.
3. Fig.2. RNA-seq should be deposited in a public repository and readily accessible. (Fig.2.d,p) the time of infection should be indicated. If overexpression of DTX3L reduces the FPN1 protein level in Fig.2e, h and Fig.S2a, it is not clear why this effect disappears in Fig. 2m, n and Fig.S2q.
4. Line 131-133 and Fig. S2h-p, the authors claim that viruses activate IFNAR-JAK signaling to increase DTX3L expression and subsequently promote degradation of FPN1. However, the authors also show that IFN-13 has no impact on the FPN1 protein level in Fig. 1i and Line 100-102. If IFN-13 can increase DTX3L expression via IFNAR-JAK signaling, why does this not lead to FPN1 degradation?
5. In Fig.S2f, g, at 1 h post virus infection, the Dtx3l mRNA seems not upregulated while its protein level increases. Please explain.
6. Fig.3. a,b,c,d,g,j,m,n,q: the time of infection should be indicated. HSV-1/VSV replicates poorly and slowly in macrophages. It is surprising to see productive replication in just a few hours (presumably 8 hr). It is not clear how infection was performed. The MOI should be provided, and the initial inoculum should also be titrated to compare with the titers in Fig.3a, b.
7. Fig.4f., time of infection is missing.
8. Fig.5. a,b,h,n: the time of infection is missing. Again, like Fig.3a,b more experimental information is required to ascertain productive viral replication. DFO enhances IFN-13 expression in WT cells (Fig.5.i,j), and this correlates with reduced VSV replication (Fig.5.l), but HSV replication is not affected (Fig.5.k).
9. Fig.6. "interperitoneally" or "intraperitoneally"? a,b,d,e,f,g) what is the time point (day) after infection? In Fig.6. i) the summary figure should be placed in Fig.7.
10. Fig.7. Are the doses of VSV used in Fig.6d and Fig.7l the same? Since all these mice are on the C57BL/6 background, why are the survival rates of the wildtype mice in the two experiments are so different if the same dose is used? Again, the MOI and time of infection/ligand stimulation should be specified.
11. Please check all the supplemental figures for missing MOI/time.
12. Fig. S1b, please indicate what does y-axis (count) represent (count cell number?), and what do the red and blue color represent?
13. Line 113/123: "Expression" is not appropriate in these contexts. Viruses and DTX3L regulates FPN1 protein degradation/stability, not expression.
14. Line 152" remove "to" from "to against".
15. Line239, "Similarly, Dtx3l knockdown attenuated RLR activation (Fig. S7e-h)." I think Dtx3l knockdown enhanced RLR activation.
16. Line379-386, details are needed for the QIP assay even though a reference is provided.
17. Line381, FeHQ seems redundant.

Reviewer #3

(Remarks to the Author)

Reviewer #4

(Remarks to the Author)

In this manuscript, Tong et al. investigated the role of iron in innate immune responses and viral manipulations of iron levels in infected cells. The authors discovered that VSV and HSV-1 infections induce the expression of DTX3L, which causes polyubiquitination and degradation of FPN1 leading to an increase of iron. Importantly, excess amount of ferrous suppresses host antiviral response through inducing TBK1 hydroxylation and STING carbonylation, as well as inhibits autophagy. By in vivo infection models, the authors further demonstrate the physiological functions of FPN1 and DTX3L in iron homeostasis and innate immune responses. Overall, the authors have provided comprehensive and high-quality data to support their working model. While the conclusions are largely convincing, I have several comments and suggestions that are listed as follows:

(1) Based on Figure 2, it seems that DTX3L interacts with FPN1 regardless of viral infections. The authors should test whether viral infections promote the interaction between DTX3L and FPN1, or merely induce the protein level of DTX3L. It is hard to draw conclusions on the binding affinity based on Figures 2i and 2j as FPN1 was degraded, which can be overcome by including MG132. Additionally, the authors should check the protein level of an irrelevant substrate of DTX3L to test whether the increased DTX3L during viral infections specifically targets FPN1.

(2) In Figure 4a, the overall fold of IRF3-5D was quite low (as compared to the >50 fold in Figure 5m), which undermines the conclusion that Fe²⁺ targets TBK-1 but not further downstream. The authors should repeat this assay and further complement it by including IRF3-5D in Figure 4b.

(3) Figures 5f and 5g, LC3-II/actin failed to decrease in CKO cells in the absence of BafA1, in start contrast to the evident increase in DFO-treated cells without BafA1 (Figures 3t and 3u). The authors should provide explanation for this discrepancy.

(4) Please provide the exact titers of VSV used for mice infections as presented in Figures 6d and 7a. The survival rate of WT and Dtx3l^{+/+} (both being C56BL/6 background) differed significantly yet there was only viral titer provided in the materials and methods.

(5) Line 202, 'RANTES' instead of 'RNATES'.

(6) Please provide more discussion on the pro-type-I interferon roles of DTX3L as other studies have demonstrated multiple mechanisms underlying its functions in promoting innate immune signaling.

(7) Line 300, FPN 1-mediated iron homeostasis may not be required for the 'initiation' of PRR signaling, but its function seems to target downstream signaling event.

Version 1:

Reviewer comments:

Reviewer #2

(Remarks to the Author)

I am satisfied with the authors' response on addressing my concerns and revised manuscript.

Reviewer #3

(Remarks to the Author)

Reviewer #4

(Remarks to the Author)

The revised manuscript has adequately addressed all my concerns by providing revised data and new discussion. I have no more questions.

Reviewer #5

(Remarks to the Author)

To the authors:

This manuscript by Li Tong et al. describe a series of alterations in cell signalling events that the authors claim to show an interference by viruses in ferroportin turnover and the activation of anti-viral host defense.

In my opinion, the evidence presented in the manuscript is not enough to reach the claimed conclusions, namely because there are several aspects of the methodology and experimental models that are not clearly explained.

1. Regarding the content of the introduction: Lines 54 to 64 in the introduction deal with bacteria and fungi and are not relevant for this study. Conversely, given the new findings of this work, it would be useful to have in the introduction some background on what is known about the ubiquitination of ferroportin and role of hepcidin.

2. Key aspects of the methods and experimental models are missing (e.g. number of cells, type of plastic support- flasks, well plates, etc.- to assist in understanding the amount of biological material analysed).

I could not find at any point the description of the infection of the cells with the viruses. It is not clear whether there is an infection or not, i.e., whether viral particles enter the cells and replicate or cells are just transfected with viral DNA. In line 422 (Methods section "RNA-Seq experiment and analysis"), it is said "After HSV-1, VSV, or SeV STIMULATION, total RNA was extracted from peritoneal macrophages" and in line 470 (Methods section "Immunofluorescence staining and confocal

analysis") it is said "Macrophages were STIMULATION with HSV-1 or VSV." In line 475, we can read "The PMs WERE TREATED with the virus, and the solution was subsequently diluted". Conversely, in line 204 (results section) it is said "attenuated HSV-1-, VSV-, SeV-, ISD-, poly(I:C) transfection-, and cGAMP transfection -induced expression". While cGAMP transfection seems to be just a mistake, the former items (HSV, VSV...) suggest that the authors actually did transfections all the time, and not infections with whole viruses.

There are no measurements of viral replication or (even more relevant) of the viability of the infected cells.

The procedures for liver macrophage isolation (a complex procedure), cell transfection and immunoprecipitation are also not described, as far as I could see.

3.1- I have several doubts regarding the way the authors measured intracellular free iron (Figure 1, a, b). First, the so called "quenchable iron pool (QIP)" is determined by a very indirect method, that actually measures the amount of iron internalized by a cell in a certain time frame. In the original publication of the method, the authors claim this is a measure of the "iron-starving state" and that the cell takes up as much iron as it is in need. This is highly questionable; however, in that manuscript, they complement these measurements with other indirect determinations of intracellular iron, such as IRP capacity to bind IRE. In the present manuscript, Li Tong et al. do not add any other measurements and (in my opinion, abusively) use the QIP as synonymous to "intracellular free iron". In fact, the decreased uptake of iron after viral infection/exposure could result from alterations in iron-related proteins such as TfR or DMT-1, and not from an increased intracellular free cytosolic iron pool .

Furthermore, the method implies the preparation of a compound called FeQH, from a combination of iron chloride with 8-hydroxyquinoline. I noticed that the later reagent is not part of the list of reagents described in the methods section. (Instead, there is an 3-ethynylaniline (m-APA; 498289), the usage of which is not explained). The authors do not describe the preparation of FeQH, although they state in lines 390-391 "The FeHQ complex optimized according to the existing literature was used".

3.2. The fact that the authors describe intracellular iron quantification by "Enzyme-linked immunosorbent assay (ELISA)" and include it in the list of ELISA assays (lines 436-437) is very worrisome, as the method has nothing to do with ELISA. Even more when they respond to a previous concern from Reviewer #1 with the claim of having differentially quantified Fe³⁺ by the same method. In the text, line 417 of materials and methods, the authors say that a probe was used, which is also incorrect, as the method is based in a chemical reaction occurring in cell extracts and not in the binding or distribution of a probe inside living cells. On the other hand, in the methods section there is a reference for a colorimetric method. This is a real iron quantification method. However, in my experience, these type of methods have a very low sensitivity and we need to use huge numbers of cells to get a signal. The supplier of the kit states a sensitivity of 8 micromolar, so the authors should explain how they achieved the values of the nanomol order of magnitude (Figure 1, d, l). Additionally, the way iron quantification is shown in the graph does not correspond to the description in Materials and Methods. The authors say that the results were "normalized to the total number of cells", but in the graph says Fe (nmol). It is impossible to understand this quantity- is it per cell? Per million cells? Per mL of cell extract?

4. Regarding the expression of the various proteins involved in iron regulation, the authors show western-blot images. They do not describe in the methods section the way they quantified the intensity of the bands, although they say "The values were normalized to those of actin" and some numbers appear in the image, but only for FPN1, not for other proteins. In Figure 1, e-h - The images of the western-blot do not exclude the existence of differences in the expression of proteins other than FPN1. Fluctuations in the intensity of other bands are apparent. A quantification of the bands should be performed to be able to state that only FPN1 is altered. The quantification of the other surface proteins (namely TfR) by cytometry should have also been performed, to have a more complete picture of iron-related alterations.

5. Several passages are very confusing and/or scientifically incorrect.

Line 108 "FPN1 deficiency abolished the decrease in total cellular iron and ferrous levels..."- and line 110 "viruses disrupt iron withholding by suppressing the host FPN1 expression" : besides being questionable as interpretation of the data, these sentences make no sense per se. In the first case, I guess the authors meant "FPN1 deficiency abolished the increase in total cellular iron and ferrous iron levels...". In the second case, "viruses disrupt iron release, by suppressing the host FPN1 expression".

6. Figure 2d- The way the RNA-seq results are presented is not very obvious. A Venn diagram would be more intuitive. The legend of the figure states "Significantly enriched biological processes of the E3 ligase as RNA...", which is a very confusing way of describing the results. Additionally, the way the RNA-seq method is described is not clear, particularly lines 430-433: "Genes with a fold change (FC) of >4 and an FDR of <0.001 after three treatments with HSV-1, VSV, and SeV were selected for further study. The identified differences were intersected with the E3 gene to obtain E3 with an FC of >4 and an FDR of <0.001. Subsequently, a schematic diagram was generated."

7. The term "ferrous" is often used instead of "iron". For example: Line 24 (in the abstract) and line 71 (in the introduction) " ferrous suppresses type I IFN" should read "ferrous iron (or simply iron) suppresses type I IFN". Line 93: "accumulation of ferrous in PMs"

8. It would be important to have the exact number (or verifiable reference) of the licence(s) obtained to perform all the procedures involving animals.

Version 3:

Reviewer comments:

Reviewer #5

(Remarks to the Author)

Reviewer #6

(Remarks to the Author)

I've been asked to comment on the revisions performed to address Reviewer #5's concerns. The responses all appear adequate. The inconsistencies and lack of methodological details noted by the reviewer have been addressed.

Reviewer #1:

Zhao and colleagues describe a novel mechanism of viral escape to immune response at the cellular levels by modulating iron levels. Viral infection leads to the upregulation of DTX3L that in turn, induced the degradation of ferroportin, causing intracellular iron accumulation. Excessive Fe²⁺ levels reduced type I IFN responses. The manuscript is clearly organized and well written. Nevertheless, I have two major concerns:

Answer: Thank you for your thorough review of our manuscript and insightful comments. We have carefully revised the manuscript and performed additional experiments according to your valuable suggestions, and in doing so we have strengthened the mechanistic details and physiological relevance of the findings.

1) Intracellular iron is mostly in the Fe³⁺ form (bound to ferritin) as the labile iron pool (Fe²⁺) must be kept minimal to avoid cellular damage. The study addressed the effects of Fe²⁺ ions but did not study the effect of infection on ferritin levels and Fe³⁺ levels.

Answer: Thank you for your comments and suggestion. We have performed additional experiments to test ferritin levels and free Fe³⁺ levels after viral infection according to your suggestion. There were no changes of ferritin levels after both HSV-1 and VSV infection in PMs and THP-1 cells. The free Fe³⁺ levels increased slightly at 8 h after HSV-1 and VSV infection.

Figure S1m

Figure S1n

Response Figure 1

Response Fig.1 ELISA analysis of Fe³⁺ in mouse PMs infected with HSV-1 (MOI:10) or VSV (MOI:1) for the indicated time periods.

These results revealed that Fe^{3+} , especially ferritin bound form, were much more stable after viral infection than Fe^{2+} , which indicated that Fe^{2+} was more labile and likely to affect innate immune responses. We added the ferritin levels in the revised manuscript. The data of Fe^{3+} do not affect the conclusion in this manuscript, so we did not include this data in the revised manuscript.

2) The authors resorted to *in vivo* models of viral infection to confirm their *in vitro* findings. However, when looking to iron metabolism/ homeostasis at the systemic level, hepcidin-mediated effects should be considered. Hepcidin is upregulated during inflammatory conditions, such as viral infections, and binds to FPN1 leading to its degradation. Thus, hepcidin levels and its effects on FPN1 levels should be determined to dissect the relevance of the DTX3L-mediated FPN1 degradation in the *in vivo* models of viral infection.

Answer: Thank you for your valuable suggestion. For *in vitro* models, we detected the expression of hepcidin after viral infection in PMs. The expression of hepcidin kept unchanged in rest state and 1 h after viral infection (Response Fig. 2a, b), during which the expression of FPN1 has already significantly decreased (Fig. 1e, f), indicated that hepcidin was not the main factor inducing the degradation of FPN1.

Response Figure 2

Response Fig.2 a) ELISA analysis of hepcidin in mouse PMs infected with HSV-1 (MOI: 10) or VSV

(MOI: 1). b) RT-PCR analysis of the mRNA expression of *hepcidin* gene in PMs infected with HSV-1 (MOI: 10) or VSV (MOI: 1) for the indicated time periods. c) ELISA analysis of hepcidin in mouse infected with HSV-1 (2×10^7 PFU/mouse) for 4 h or VSV (5×10^8 PFU/mouse) for 8 h. d) RT-PCR analysis of the mRNA expression of *hepcidin* gene in PMs for 4 h. e) Western blot analysis of FPN1 and DTX3L in PMs from *Dtx3l*^{+/+} or *Dtx3l*^{-/-} mice, pretreated with hepcidin inhibitor, followed by infection with HSV-1 (MOI: 10) for 4 h or VSV (MOI: 1) for 8 h.

For *in vivo* models, the hepcidin levels in the serum of HSV-1 and VSV infected mouse model increased slightly (Response Fig. 2c), also indicated that hepcidin was not the main factor inducing FPN1 degradation. In addition, we detected the influence of DTX3L in the presence of hepcidin inhibitor (Response Fig. 2d). The degradation of FPN1 was not affected by HI treatment after viral infection, which could be abolished in *Dtx3l* deficiency (Response Fig. 2e). However, we did not performance this experiment *in vivo* because the effect of DTX3L could not target macrophages specifically by using *Dtx3l* systemic knockout mice.

Therefore, these results indicated that DTX3L was the main factor inducing FPN1 degradation both *in vitro* and *in vivo* in the early stage of viral infection. These data do not affect the conclusion in our manuscript, so we did not include these data in the revised manuscript.

Minor concerns:

1. Line 93 “These findings indicate that viruses disrupt host iron withholding and cause iron accumulation in the cells.” There is no evidence in the results to support the conclusion that viruses disrupt host iron withholding at this point.

Answer: Thank you for the valuable suggestion; we have replaced this sentence by “These findings indicate that viral infection induces the ferrous iron accumulation in the cells.”

2. Line 97 “a significant downregulation of the sole cellular iron exporter, FPN1 ”. Have the authors performed a quantitative analysis followed by statistical testing to support this statement?

Answer: We are very sorry for the inaccurate statement. The repeat experiments of proteins expression levels in Fig. 1e, 1f, S1i, and S1j were quantitated by measuring band intensities using ‘ImageJ’ software

and the values were normalized to actin. Considering that these time point data of Western blots are not suitable for statistical analysis, we revised the sentence in revised manuscript: Our findings revealed that only the sole cellular iron exporter, FPN1, was downregulated after viral infection in mouse PMs and human THP-1 cells. We added the quantitative data of band intensities in the revised manuscript.

Along this line, the amount of FPN1 protein at 12h post-IFN β stimulation (Fig 1i) is similar to that observed for the same timepoint but for viral infections (Fig 1e, 1f), and the authors consider that IFN β has not direct effect on FPN1 levels. Moreover, the kinetics of FPN1 downregulation seem to differ between a primary cell (mouse PM), and a cell line (THP-1). I would like the authors to give a rationale for the difference.

Answer: The secreted IFN-I3 activates JAK-STAT pathway through IFNAR, and then induces a large amount of genes expression. In our experiments, we showed that IFN-I3 has no effect on promoting the expression of FPN1. On the contrary, the FPN1 protein level slightly decreased after 12 h of IFN-I3 stimulation. Moreover, the expression of DTX3L increased after IFN-I3 stimulation, which in turn promoted the degradation of FPN1 at later time point after IFN-I3 stimulation.

The FPN1 protein levels decreased after viral infection both in mouse PMs and THP-1 cells. The only difference between them was the timing of the FPN1 decline. Rather than primary mouse PMs,

THP1 cells are induced macrophages by PMA, which may be less sensitive to viral infection and be slightly delayed of FPN1 downregulation.

1. Lines 111-113 “Notably, this effect was reversed by the proteasome inhibitor MG132, but not by the lysosome inhibitors chloroquine and 3methyladenine(3-MA) (Fig.2c).” The data do not depict what is described in those lines, namely regarding chloroquine effect on FPN1 levels.

Answer: Thanks for your comments. In Fig.2c, the degradation of FPN1 was blocked by the proteasome inhibitor MG132, indicated that the degradation of FPN1 is mainly through the proteasome pathway. While the expression of FPN1 kept unchanged no matter treated with the lysosome inhibitors chloroquine/3-MA or not, indicated that the lysosomal pathway is not involved in the degradation process of FPN1. We rewrote this description in the revised manuscript.

2. Lines 130-134: Does the treatment with NF- κ B inhibitors (JSH-23 and QNZ) and JAK1 inhibitor itacitinib affect FPN1 levels?

Answer: Thanks for the valuable question. FPN1 levels were not affected by NF- κ B inhibitors (JSH-23 and QNZ) nor JAK1 inhibitor itacitinib after viral infection. We added these new data in the revised manuscript.

Figure S1o

Figure S1p

Figure S1q

Reviewer #2:

Cells maintain iron homeostasis for normal cellular physiology and functions. Viruses are known to hijack iron metabolism to facilitate their replication. In this manuscript, Li Tong and colleagues report a negative role of iron in antiviral innate immunity. The authors show that either DNA or RNA virus infections induce polyubiquitination and degradation of ferroportin (FPN1), leading to intracellular iron

accumulation rapidly. FPN1 ubiquitination and degradation is mediated by a host E3 ligase, DTX3L, which expression is upregulated following viral infection. DTX3L interacts with FPN1 and mediates its K48-linked polyubiquitination. Excessive iron suppresses type I IFN responses and autophagy by promoting TBK1 hydroxylation and STING carbonylation. Therefore, FPN1 deficiency results in impaired innate immunity while enhanced viral replication in vitro and in vivo. In contrast, DTX3L deficiency has the opposite effect. The study is comprehensive; the results are interesting and novel.

Answer: Thank you for your thorough review of our manuscript. We have carefully revised the manuscript and performed several additional experiments according to your valuable and insightful suggestions, and in doing so we have strengthened the mechanistic details and physiological relevance of the findings.

Specific comments:

1. In all the viral infection/ligand stimulation experiments/figures, the MOI and time of infection/stimulation should be specified in their corresponding figure legends (including supplemental figures). “US” should be spelled out in each figure legend.

Answer: Thanks for your valuable advice. All the required information was added in the revised manuscript.

2. Fig.1g,h, l: the time of infection/ligand stimulation should be indicated. There are two DMT1 bands in Fig. 1e, f, but one band in Fig. 1i and Fig. S1i and j. Please clarify this discrepancy and indicate the correct band.

Answer: Thanks for your valuable advice. We added the time of infection and ligand stimulation in the legends of revised manuscript.

We used two different batches antibodies (sc-166884 purchased from Santa Cruz Biotechnology) to detect the DMT1 molecule and one batch exhibited two bands (Fig.1e, f). We have added arrows to indicate DMT1 in the revised manuscript.

3. Fig.2. RNA-seq should be deposited in a public repository and readily accessible. (Fig.2.d,p) the time of infection should be indicated. If overexpression of DTX3L reduces the FPN1 protein level in Fig.2e, h and Fig.S2a, it is not clear why this effect disappears in Fig. 2m, n and Fig.S2q.

Answer: Thanks for your valuable advice. We submitted the RNA-seq data to Gene Expression Omnibus (GEO) Database (<http://www.ncbi.nlm.nih.gov/geo/>) under super series number GSE280144. Because the number remains in private status (<https://www.ncbi.nlm.nih.gov/geo/query/acc.cgi?acc=GSE280144>), the secure token of GSE280144 is as follows: mvixawcopzejdur.

We also added the time of infection in the legends of revised manuscript.

The expression levels of FPN1 with ubiquitination chains would be largely reduced without MG132. So MG132 was added into all the experiments associated ubiquitination comparison (including Fig.2m, n and S2q), which we annotated in the legends.

4. Line 131-133 and Fig. S2h-p, the authors claim that viruses activate IFNAR-JAK signaling to increase DTX3L expression and subsequently promote degradation of FPN1. However, the authors also show that IFN- β has no impact on the FPN1 protein level in Fig. 1i and Line 100-102. If IFN- β can increase DTX3L expression via IFNAR-JAK signaling, why does this not lead to FPN1 degradation?

Answer: Yes, it is an important question. The secreted IFN- β activates JAK-STAT pathway through IFNAR, and then induces a large amount of genes expression. In our experiments, we showed that IFN- β has no effect on inducing the production of FPN1. So the description “Notably, FPN1 expression remained unchanged after IFN- β stimulation” is not precise. We revised this sentence as “Notably, IFN- β has no effect on inducing the production of FPN1”. Indeed, the FPN1 protein level decreased at 12 h after IFN- β stimulation (Fig. 1i), which is correlated with the result that IFN- β promoted the expression of DTX3L.

3. In Fig.S2f, g, at 1 h post virus infection, the *Dtx3l* mRNA seems not upregulated while its protein level increases. Please explain.

Answer: Thank you for your question. The *Dtx3l* mRNA and protein levels after viral infection in mouse PMs was affected by individual differences and viral virulence, so there would be slight differences in the trend of each experiment. We performed the experiment in Fig. S2f and S2g again by using the same mouse and viruses to make mRNA consistent with protein levels. We replaced this data in the revised manuscript.

Figure S2f

Figure S2g

5. Fig.3. a,b,c,d,g,j,m,n,q: the time of infection should be indicated. HSV-1/VSV replicates poorly and slowly in macrophages. It is surprising to see productive replication in just a few hours (presumably 8 hr). It is not clear how infection was performed. The MOI should be provided, and the initial inoculum should also be titrated to compare with the titers in Fig.3a, b.

Answer: Thank you for your valuable advice. We added the time of infection in Fig.3 in the revised manuscript.

For Fig. 3a, b, mouse PMs pretreated with 10 nM Fe²⁺, followed by infection with HSV-1 (MOI:10) for 8 h or VSV (MOI:1) for 12 h. Then the cell culture supernatants were collected to perform the 50% tissue culture infectious dose assay. The time of infection and MOI we used were referring to published literature (J Immunol. 2021;207(7):1903-1910.). Furthermore, the viruses productive replication could indeed be observed in our previously and other laboratories studies (Immunity. 2023;56(11):25082522.e6.; Cell Mol Immunol. 2021;18(10):2358-2371).

4. Fig.4f., time of infection is missing.

Answer: We are sorry for it and have added the time of infection in Fig.4f in the revised manuscript.

6. Fig.5. a,b,h,n: the time of infection is missing. Again, like Fig.3a,b more experimental information is required to ascertain productive viral replication. DFO enhances IFN- β expression in WT cells (Fig.5.i,j), and this correlates with reduced VSV replication (Fig.5.l), but HSV replication is not affected (Fig.5.k).

Answer: Thank you for your valuable advice. We added more information of Fig.3 and Fig. 5 in the revised manuscript.

For Fig. 5k, l, the experiments were performed again and revealed that the treatment of DFO

inhibited VSV and HSV replication, which correlated with the IFN- β expression. We added these data in the revised manuscript.

7. Fig.6. “interperitoneally” or “intraperitoneally”? a,b,d,e,f,g) what is the time point (day) after infection? In Fig.6. i) the summary figure should be placed in Fig.7.

Answer: We are sorry for this mistake. We corrected it as “intraperitoneally” in the revised manuscript. For ELISA, serum was collected 12 h after VSV infection or 8 h after HSV-1 infection. For RT-PCR and HE staining, tissues were collected 36 h after VSV infection or 24 h after HSV-1 infection. We added the information in the revised manuscript.

According to your suggestion, we have rearranged the summary figure from Fig.6i to Fig.7.

8. Fig.7. Are the doses of VSV used in Fig.6d and Fig.7i the same? Since all these mice are on the C57BL/6 background, why are the survival rates of the wildtype mice in the two experiments are so different if the same dose is used? Again, the MOI and time of infection/ligand stimulation should be specified.

Answer: Thank you for your valuable advice. The titers of VSV used in Fig.7i was 5×10^8 PFU/mouse. Then we performed the experiment in Fig.6d used the same titers and found that almost all of the WT and CKO mice died in the early stages of viral infection. So we adjusted the virus titer to 5×10^7 PFU/mouse in Fig. 6d. We added this information in the revised manuscript.

9. Please check all the supplemental figures for missing MOI/time.

Answer: Thank you for your valuable advice. We added the missing MOI/time in the supplemental figures.

10. Fig. S1b, please indicate what does y-axis (count) represent (count cell number?), and what do the red and blue color represent?

Answer: Thank you for your valuable advice. The MFI (calcein) was determined in naïve (blue) and FeHQ-challenged (red) PMs, and the QIP was calculated as the difference in MFI. We added this information and indicated the Y-axis in the revised manuscript.

Figure S1b

11. Line 113/123: “Expression” is not appropriate in these contexts. Viruses and DTX3L regulates FPN1 protein degradation/stability, not expression.

Answer: Thank you for your valuable advice. Viruses promote FPN1 degradation through proteasome pathway. We rewrote these contexts in the revised manuscript.

12. Line 152” remove “to” from “to against”.

Answer: Thank you for your carefully review of our manuscript. We corrected it as the suggestion in the revised manuscript.

13. Line239, “Similarly, Dtx3l knockdown attenuated RLR activation (Fig. S7e-h).” I think Dtx3l knockdown enhanced RLR activation.

Answer: We are sorry for this mistake. We corrected it as the suggestion in the revised manuscript.

14. Line379-386, details are needed for the QIP assay even though a reference is provided.

Answer: Thank you for your valuable advice. We added more details about QIP assay in the revised manuscript.

PMs were washed 3x with PBS and stained with 300 μ L of 1 μ M Calcein-AM (BioLegend) in PBS for 15 min at 37°C. After two PBS washing steps, PMs were left untreated in 300 μ L PBS or were treated

with 300 μ L FeHQ solution for 30 min at 37°C. The FeHQ complex optimized according to the existing literature was used. The Calcein median fluorescence intensity (MFI) of the PMs was collected on a CytoFLEX (Beckman) and analyzed using FlowJo software version 10 (FlowJo). The QIP of PMs per experimental condition was calculated as the difference between the MFI (untreated) and MFI (FeHQ treated). At least three technical replicates were analyzed under the untreated and FeHQ-treated conditions.

17. Line381, FeHQ seems redundant.

Answer: Thank you very much for your review of our manuscript carefully. We deleted the redundant “FeHQ” in the revised manuscript.

Reviewer #3 :

Answer: Thank you for your thorough review of our manuscript. We have carefully revised the manuscript and performed additional experiments according to your valuable suggestions, and in doing so we have strengthened the mechanistic details and physiological relevance of the findings.

Reviewer #4:

In this manuscript, Tong et al. investigated the role of iron in innate immune responses and viral manipulations of iron levels in infected cells. The authors discovered that VSV and HSV-1 infections induce the expression of DTX3L, which causes polyubiquitination and degradation of FPN1 leading to an increase of iron. Importantly, excess amount of ferrous suppresses host antiviral response through inducing TBK1 hydroxylation and STING carbonylation, as well as inhibits autophagy. By in vivo infection models, the authors further demonstrate the physiological functions of FPN1 and DTX3L in

iron homeostasis and innate immune responses. Overall, the authors have provided comprehensive and high-quality data to support their working model. While the conclusions are largely convincing, I have several comments and suggestions that are listed as follows:

Answer: Thank you for your thorough review of our manuscript. We have carefully revised the manuscript and performed additional experiments according to your valuable and insightful suggestions, and in doing so we have strengthened the mechanistic details and physiological relevance of the findings. Please find below our responses to the comments and suggestions.

(1) Based on Figure 2, it seems that DTX3L interacts with FPN1 regardless of viral infections. The authors should test whether viral infections promote the interaction between DTX3L and FPN1, or merely induce the protein level of DTX3L. It is hard to draw conclusions on the binding affinity based on Figures 2i and 2j as FPN1 was degraded, which can be overcome by including MG132. Additionally, the authors should check the protein level of an irrelevant substrate of DTX3L to test whether the increased DTX3L during viral infections specifically targets FPN1.

Answer: Thank you for your valuable suggestions. According to your advice, we performed the IP experiments by using the proteasome inhibitor MG132 and the protein synthesis inhibitor cycloheximide. Indeed, when we blocked the degradation of FPN1 by using MG132 or blocked the synthesis of DTX3L by using cycloheximide, the interaction between DTX3L and FPN1 was both promoted by VSV and HSV-1 infection. These results indicated that viral infections promote the interaction between DTX3L and FPN1 (Response Figure 3). These data do not affect the conclusion in our manuscript, so we did not include these data in the revised manuscript.

Furthermore, we also showed that the DMT1 protein levels were not affected in *Dtx3l*^{-/-} mouse PMs (Figure 2f), indicated that the increased DTX3L during viral infections specifically targeted FPN1.

Response Fig.3 a) Lysates from mouse PMs pretreated with MG132 (10 μ M) infected with HSV-1 (MOI:10) for 4 h or VSV (MOI:1) for 8 h, then immunoprecipitated with FPN1 antibody, followed by Western blot analysis with indicated antibodies. b) Lysates from mouse PMs pretreated with CHX (10 μ M) infected with HSV-1 (MOI:10) for 4 h or VSV (MOI:1) for 8 h, then immunoprecipitated with DTX3L antibody, followed by Western blot analysis with indicated antibodies.

(2) In Figure 4a, the overall fold of IRF3-5D was quite low (as compared to the >50 fold in Figure 5m), which undermines the conclusion that Fe²⁺ targets TBK-1 but not further downstream. The authors should repeat this assay and further complement it by including IRF3-5D in Figure 4b.

Answer: Thank you for your valuable suggestion. We repeated this experiment in Fig.4a and replaced it in the revised manuscript. However, it is only adaptor which upstream of IRF3 could regulate IRF3 gene reporter assay, so we did not detect IRF3-5D induced IRF3 lucif. activation.

(3) Figures 5f and 5g, LC3-II/actin failed to decrease in CKO cells in the absence of BafA1, in start contrast to the evident increase in DFO-treated cells without BafA1 (Figures 3t and 3u). The authors should provide explanation for this discrepancy.

Answer: We are sorry for this mistake. We chose the wrong band when organizing the data. We replaced the right figures in the revised manuscript.

(4) Please provide the exact titers of VSV used for mice infections as presented in Figures 6d and 7a. The survival rate of WT and Dtx3l^{+/+} (both being C56BL/6 background) differed significantly yet there was only viral titer provided in the materials and methods.

Answer: Thank you for your valuable advice. The titers of VSV used in Fig.7l was 5×10^8 PFU/mouse. Then we performed the experiment in Fig.6d used the same titers and found that almost all of the WT and CKO mice died in the early stages of viral infection. So we adjusted the virus titer to 5×10^7 PFU/mouse in Fig. 6d. We added this information in the revised manuscript.

(5) Line 202, 'RANTES' instead of 'RNATES'.

Answer: We are so sorry for this mistake. We corrected it as the suggestion in the revised manuscript.

(6) Please provide more discussion on the pro-type-I interferon roles of DTX3L as other studies have demonstrated multiple mechanisms underlying its functions in promoting innate immune signaling.

Answer: Thank you for your valuable suggestion. As an important E3 ligase in DNA damage response (*Mol Cell Biol.* 2013; 33:845–57), DTX3L have been recently implicated in enhancing interferon signaling by mediating histone H2BJ monoubiquitination (*Nat Immunol.* 2015;16(12):1215-27) and TBK1 K63-linked ubiquitination (*J Virol.* 2023;97(6):e0068723). In our study, DTX3L selectively facilitates the K48-linked ubiquitination and protein degradation of FPN1, thus disrupting host iron withholding during viral infection. These different results indicated that different types of ubiquitination modifications, different cell types and viruses led to different functions of DTX3L. We added these contents in the discussion of revised manuscript.

(7) Line 300, FPN1-mediated iron homeostasis may not be required for the 'initiation' of PRR signaling,

but its function seems to target downstream signaling event.

Answer: We are so sorry for the inaccurate description. Our results indicate that FPN1 -maintained intracellular iron homeostasis is required for the manipulation of cGAS- and RLR-dependent downstream innate responses, revealing an additional immune escape mechanism and thereby providing promising therapeutic targets for the treatment of viral diseases. We rewrote this sentence in the revised manuscript.

Reviewer #2 :

I am satisfied with the authors' response on addressing my concerns and revised manuscript.

Answer: We appreciate very much for your work in reviewing our manuscript. The insightful suggestions and comments help us to greatly improve the manuscript.

Reviewer #3:

Answer: We appreciate very much for your work in reviewing our manuscript. The insightful suggestions and comments help us to greatly improve the manuscript.

Reviewer #4:

The revised manuscript has adequately addressed all my concerns by providing revised data and new discussion. I have no more questions.

Answer: We appreciate very much for your work in reviewing our manuscript. The insightful suggestions and comments help us to greatly improve the manuscript.

Reviewer #5:

This manuscript by Li Tong et al. describe a series of alterations in cell signalling events that the authors claim to show an interference by viruses in ferroportin turnover and the activation of anti-viral host defense. In my opinion, the evidence presented in the manuscript is not enough

to reach the claimed conclusions, namely because there are several aspects of the methodology and experimental models that are not clearly explained.

Answer: Thank you for your thorough review of our manuscript and insightful comments. We have carefully revised the manuscript and performed additional experiments according to your valuable suggestions, and in doing so we have strengthened the mechanistic details and physiological relevance of the findings.

1. Regarding the content of the introduction: Lines 54 to 64 in the introduction deal with bacteria and fungi and are not relevant for this study. Conversely, given the new findings of this work, it would be useful to have in the introduction some background on what is known about the ubiquitination of ferroportin and role of hepcidin.

Answer: Thank you for valuable advice. The information of the ubiquitination of FPN1 and role of hepcidin is important so we added relevant descriptions in the revised version: As the only known cellular iron exporter, FPN1 plays a central role in intracellular iron homeostasis, so the FPN1 levels need to be tightly regulated. Hepcidin induces the endocytosis and ubiquitination of FPN1, followed by degradation predominantly in lysosomes. E3 ubiquitin ligase RNF217 mediates the ubiquitination and degradation of FPN1.

The main idea of this manuscript lies on the iron withholding of viruses. It is necessary to introduce the iron withholding of bacteria and fungi, so we kept relevant descriptions in the revised manuscript.

2. Key aspects of the methods and experimental models are missing (e.g. number of cells, type of plastic support- flasks, well plates, etc.- to assist in understanding the amount of biological material analysed).

Answer: Thank you for your valuable suggestions. We checked the entire manuscript and added more details of the methods and experimental models in the revised manuscript.

I could not find at any point the description of the infection of the cells with the viruses. It is not clear whether there is an infection or not, i.e., whether viral particles enter the cells and replicate or cells are just transfected with viral DNA. In line 422 (Methods section “RNA-Seq experiment and analysis”), it is said “After HSV-1, VSV, or SeV STIMULATION, total RNA was extracted from peritoneal macrophages” and in line 470 (Methods section “Immunofluorescence staining and confocal analysis”) it is said “Macrophages were STIMULATION with HSV-1 or VSV.” In line 475, we can read “The PMs WERE TREATED with the virus, and the solution was subsequently diluted”. Conversely, in line 204 (results section) it is said “attenuated HSV-1-, VSV-, SeV-, ISD-, poly(I:C) transfection-, and cGAMP transfection -induced expression”. While cGAMP transfection seems to be just a mistake, the former items (HSV, VSV...) suggest that the authors actually did transfections all the time, and not infections with whole viruses.

Answer: We are sorry for the unclear expressions. All the experiments we referred SeV, VSV and HSV-1 are infection process by whole viruses, but not viral particles or viral genomes. We also wrote the multiplicity of infection (MOI) we used in the figure legends. We changed all the descriptions “HSV-1, VSV, or SeV STIMULATION” into “HSV-1, VSV, or SeV INFECTION”

in the revised manuscript.

ISD, poly(I:C), and cGAMP are cytoplasmic ligands that recognized by cGAS, RIG-I and STING, respectively. So ISD, poly(I:C), and cGAMP were transfected into mouse PMs by using Lipofectamine 2000 in our experiments. We changed the description “HSV-1-, VSV-, SeV-, ISD-, poly(I:C) transfection-, and cGAMP transfection -induced expression” into “HSV-1, VSV, SeV infection-, and ISD, poly(I:C), cGAMP transfection- induced expression... ..”. We also added the ISD, poly(I:C), and cGAMP transfection information in the method of revised manuscript.

There are no measurements of viral replication or (even more relevant) of the viability of the infected cells.

Answer: Thank you for the valuable suggestion. Detection of HSV-1 UL30 mRNA expression and VSV gene RNA replication by RT-PCR we performed in our study is widely used to evaluate HSV-1 and VSV viral replication (Nat Immunol. 2020; 21(7):727-735; Nat Immunol. 2017;18(2):214-224; Cell Mol Immunol. 2018; 15(10): 907- 916). According to your valuable suggestions, we also examined HSV-1 and VSV replication by using plaque assay, and observed marked differences in HSV-1 and VSV replication. Consider that TCID50 we used in our previous manuscript is functional redundant, so we replaced TCID50 data with these new data (acquired by plaque assay) in the revised manuscript.

The viability of the infected cells were measured by Cell Counting Kit-8 (CCK8) assay. Ferrous iron and FPN1 deficiency decreased the cell viability with the prolongation of virus infection time, consistent with the increased replication levels of HSV-1 and VSV (Response Fig.1). The

data of cell viability do not affect the conclusion in this manuscript, so we did not include this data in the revised manuscript.

Response Fig. 1

Response Fig. 1a, b Cell Counting Kit-8 (CCK-8) analysis of cell viability of mouse PMs pretreated with LAL water (Mock), or Fe²⁺ (10 nM) for 1 h, followed by infected with HSV-1(MOI:10) or VSV (MOI:1) for the indicated time periods. Response Fig.1 c, d CCK-8 analysis of cell viability of PMs from WT or *Slc40a1*^{CKO} mice, followed by infected with HSV-1(MOI:10) or VSV (MOI:1) for the indicated time periods.

The procedures for liver macrophage isolation (a complex procedure), cell transfection and immunoprecipitation are also not described, as far as I could see.

Answer: Thank you for your valuable suggestions. We added the procedures for spleen and liver macrophage isolation, cell transfection and immunoprecipitation in the method of revised manuscript.

3.1- I have several doubts regarding the way the authors measured intracellular free iron (Figure 1, a, b). First, the so called “quenchable iron pool (QIP)” is determined by a very indirect method, that actually measures the amount of iron internalized by a cell in a certain time frame. In the original publication of the method, the authors claim this is a measure of the “iron-starving state” and that the cell takes up as much iron as it is in need. This is highly questionable.

Answer: Thank you for your comments and suggestion. Calcein-AM is a widely used Fe-specific fluorescent dye locating exclusively to the cytoplasm (Metallomics. 2015; 7(2): 21222; Blood.2005;106:3242-3250). This method is not only used for measuring “iron-starving state” , but also all the experiments associated cellular iron detection (J Adv Res. 2024:S2090-1232(24)00429-6; J Ethnopharmacol. 2023;308:116267).

Calcein-AM fluoresced in unbound form and fluorescence quenched after iron binding. Furthermore, fluorescence could be totally quenched when additional exogenous FeHQ was taken up to cells. Thereby, the differences in cellular Calcein fluorescence before and after FeHQ addition represent the quenchable iron pool (QIP) of a given cell. The description “FeHQ was taken up” is just one step in the iron detection process and has nothing to do with iron-starving state. Consider that the left panel of Fig. S1c-g which showed the the differences in cellular Calcein fluorescence before and after FeHQ addition was easily to be misunderstood as a measure of the “iron-starving state” , so we removed these panels and added a new schematic representation about QIP calculation in Fig. S1b. We also rewrote the description of method and figure legends, and redrew the schematic representation about Calcein fluorescence in Fig.S1a in the revised manuscript.

Fig. S1a

Fig. S1b

However, in that manuscript, they complement these measurements with other indirect determinations of intracellular iron, such as IRP capacity to bind IRE. In the present manuscript, Li Tong et al. do not add any other measurements and (in my opinion, abusively) use the QIP as synonymous to “intracellular free iron”.

Answer: IRP1 and IRP2 are orthologous proteins that regulate the post-transcriptional expression of key iron metabolism genes by binding to IREs, which are stem-loop RNA structures located in the UTR of target transcripts. Binding to 5' UTR of *Slc40a1* (encoded FPN1) mRNA inhibited FPN1 protein translation, while binding to 3'UTR of *Slc11a2* (encoded DMT1) and *Tfrc* mRNAs protected the transcript against the endoribonucleases Regnase-1. Thus IRP-IRE binding reduced iron export, and promoted iron uptake. At low intracellular iron levels, IRP accumulates and binds to IRE, thus to promote intracellular accumulation. On the contrary, at high intracellular iron levels, IRP is degraded and the utilization of intracellular iron increased. Collectively, the IRP capacity to bind IRE reflected intracellular iron homeostasis and affected the post-transcriptional levels of iron transporters (Nat Rev Mol Cell Biol. 2024;25(2):133-155). In that mentioned manuscript, Western blot analysis of IRP2 expression was performed to investigate whether the post-transcriptional levels of iron transporters could be affected by IRE-IRP system after IFN- β stimulation. So the aim which they performed IRP expression detection was not intracellular iron determination.

According to your valuable suggestions, we detected the expression of IRP2 as that manuscript. The expression of IRP2 kept unchanged until 8 h after viral infection, indicated that IRP2 did not affect the translation of FPN1 at early stage after viral infection (Response Fig.2). In this context, we further demonstrated that the degradation of FPN1 was mediated by E3 ligase DTX3L. Besides, the decreased of IRP2 at 12 h after viral infection reflected the accumulation of intracellular iron level (Response Fig.2), which was also consistent with our results. The data of IRP2 expression do not affect the conclusion in this manuscript, so we did not include this data in the revised manuscript.

Response Fig. 2

Western blot analysis of IRP2 in mouse PMs and THP1 cells infected with HSV-1(MOI:10) or VSV(MOI:1) for the indicated time periods. Sizes in kDa are indicated on the right. The values quantitated by measuring band intensities using the “ImageJ” software were normalized to those of actin. Mean and SEM of IRP2 relative expression are calculated of 3 independent experiments.

To further confirm the accumulation of intracellular free iron levels after viral infection, additional intracellular iron detection through flow cytometry analysis of Fe²⁺ Tracker was

performed in the revised manuscript. As expected, HSV-1 and VSV infection, but not IFN- β stimulation, promoted the accumulation of intracellular ferrous iron in PMs (Figure S1j). FPN1 deficiency abolished the increase in ferrous iron levels caused by HSV-1 and VSV infections (Figure 1l). We added these data in the revised manuscript.

Figure S1j

Figure 1l

In fact, the decreased uptake of iron after viral infection/exposure could result from alterations in iron-related proteins such as TfR or DMT-1, and not from an increased intracellular free cytosolic iron pool.

Answer: Thank you for your comments and suggestion. The decreased uptake of FeHQ meant increased cellular iron accumulation, which may result from higher expression of TFR and DMT1, or lower expression of FPN1. Western blot and flow cytometry revealed that only FPN1, but not TFR or DMT1, downregulated after viral infection (Fig. 1h, S1m). These results indicated that the decreased uptake of FeHQ after viral infection resulted from alteration in

FPN1.

Furthermore, the method implies the preparation of a compound called FeQH, from a combination of iron chloride with 8-hydroxyquinoline. I noticed that the later reagent is not part of the list of reagents described in the methods section. (Instead, there is an 3-ethynylaniline (m-APA; 498289), the usage of which is not explained). The authors do not describe the preparation of FeQH, although they state in lines 390-391 “The FeHQ complex optimized according to the existing literature was used”.

Answer: Sorry for that mistake. m-APA is a reagent used to determine STING carbonylation, referred to the method “Determination of STING carbonylation”. The information of FeHQ solution (5 μM FeCl₂ and 10 μM 8-hydroxyquinoline in PBS, all from Sigma-Aldrich) was added in the method “QIP quantification Assays” of revised manuscript.

3.2. The fact that the authors describe intracellular iron quantification by “Enzyme-linked immunosorbent assay (ELISA)” and include it in the list of ELISA assays (lines 436-437) is

very worrisome, as the method has nothing to do with ELISA.

Answer: Thank you for your valuable comments. We removed intracellular iron quantification from “ELISA assays” section. Then we added a “Iron content analysis” section including this iron quantification through colorimetric method (Fig.1d, 1m) and an additional method by flow cytometry analysis using Fe²⁺ Tracker (Fig 1l, S1j).

Figure S1j

Figure 1l

Even more when they respond to a previous concern from Reviewer #1 with the claim of having differentially quantified Fe³⁺ by the same method.

Answer: Thank you for your valuable comments. The Fe²⁺ and Fe³⁺ were both detected by using the Iron Assay Kit (Abcam, Cambridge, MA). There is Fe reducer in the kit which could reduce Fe³⁺ to Fe²⁺. So for Fe²⁺ assay, added assay buffer only. For total iron (Fe²⁺ and Fe³⁺), added iron reducer. The level of Fe³⁺ was calculated by subtracting Fe²⁺ iron from total iron. We added these information in the method of revised manuscript.

In the text, line 417 of materials and methods, the authors say that a probe was used, which is also incorrect, as the method is based in a chemical reaction occurring in cell extracts and not in the binding or distribution of a probe inside living cells.

Answer: Sorry for this inaccurate description. We replaced the description “iron probe” with “iron reaction solution” in the revised manuscript.

On the other hand, in the methods section there is a reference for a colorimetric method. This is a real iron quantification method. However, in my experience, these type of methods have a very low sensitivity and we need to use huge numbers of cells to get a signal. The supplier of the kit states a sensitivity of 8 micromolar, so the authors should explain how they achieved the values of the nanomol order of magnitude (Figure 1, d, l). Additionally, the way iron quantification is shown in the graph does not correspond to the description in Materials and Methods. The authors say that the results were “normalized to the total number of cells”, but in the graph says Fe (nmol). It is impossible to understand this quantity- is it per cell? Per million cells? Per mL of cell extract?

Answer: Thank you for your insight comments. The sensitivity of the kit is 8 micromolar, so we used a large amount of cells ($40-60 \times 10^6$ cells and extracted with 1 mL assay buffer) to cover the detection range (the raw data attached below). Consider that $\mu\text{mol/L}$ (μM) was equal to nmol/mL , and we used 1 mL assay buffer to lyse cells, so the unit of previous version were nmol. According to your advice, we normalized our results to the total number of cells in the revised manuscript (Fig.1d, 1m). The description of method in previous version was incorrect and now we rewrote it.

Figure 1d

Figure 1m

Raw date of Figure-1d

Optical Density(OD) at 593nm

	HSV-1			VSV			IFN-β		
0	0.132	0.135	0.136	0.132	0.1331	0.1356	0.132	0.121	0.141
1	0.1583	0.1593	0.16	0.1623	0.1618	0.1623	0.153	0.148	0.153
2	0.166	0.167	0.1682	0.1897	0.1865	0.1814	0.142	0.131	0.133
4	0.283	0.283	0.273	0.242	0.322	0.292	0.137	0.138	0.139
8	0.415	0.456	0.427	0.634	0.612	0.641	0.154	0.141	0.142
12	0.142	0.138	0.133	0.142	0.138	0.133	0.134	0.128	0.129

Previous results

	HSV-1			VSV			IFN-β			
0	35.4951	36.6522	37.0378	35.4951	35.9193	36.8836	35.4951	31.2525	38.9663	umol/L
1	45.6387	46.0244	46.2944	47.1815	46.9886	47.1815	43.5946	41.6661	43.5946	nmol/ml
2	48.6085	48.9942	49.4571	57.7494	56.5152	54.5482	39.352	35.1094	35.8808	
4	93.7343	93.7343	89.8774	77.921	108.776	97.2055	37.4235	37.8092	38.1949	
8	144.645	160.459	149.274	229.111	220.626	231.811	43.9803	38.9663	39.352	
12	39.352	37.8092	35.8808	39.352	37.8092	35.8808	36.2665	33.9523	34.338	

Revised results

	HSV-1			VSV			IFN-β			
0	0.59158	0.61087	0.6173	0.59158	0.59866	0.61473	0.59158	0.52087	0.64944	nmol/10 ⁶ cells
1	0.76065	0.76707	0.77157	0.78636	0.78314	0.78636	1	0.72658	0.69444	0.72658
2	0.81014	0.81657	0.82428	0.96249	0.94192	0.90914	2	0.65587	0.58516	0.59801
4	1.56224	1.56224	1.49796	1.29868	1.81294	1.62009	4	0.62373	0.63015	0.63658
8	2.41076	2.67431	2.48789	3.81852	3.6771	3.86352	8	0.733	0.64944	0.65587
12	0.65587	0.63015	0.59801	0.65587	0.63015	0.59801	12	0.60444	0.56587	0.5723

Raw date of Figure-1m

	OD at 593nm				Previous results nmol/ml				Revised results nmol/10 ⁶ cells			
WT	US	0.073	0.077	0.071	US	12.7394	14.2821	11.968	US	0.31848	0.35705	0.2992
	HSV-1	0.332	0.398	0.558	HSV-1	112.633	138.089	199.799	HSV-1	2.81583	3.45222	4.99498
	VSV	0.245	0.235	0.328	VSV	79.0781	75.2212	111.09	VSV	1.97695	1.88053	2.77726
CKO	US	0.196	0.202	0.36	US	56.3223	62.4934	123.432	US	1.40806	1.56233	3.08581
	HSV-1	0.18	0.202	0.279	HSV-1	54.0082	62.4934	92.1915	HSV-1	1.35021	1.56233	2.30479
	VSV	0.23	0.219	0.344	VSV	73.2927	69.0501	117.261	VSV	1.83232	1.72625	2.93153

4. Regarding the expression of the various proteins involved in iron regulation, the authors show western-blot images. They do not describe in the methods section the way they quantified the intensity of the bands, although they say “The values were normalized to those of actin” and some numbers appear in the image, but only for FPN1, not for other proteins. In Figure 1, e-h - The images of the western-blot do not exclude the existence of differences in the

expression of proteins other than FPN1. Fluctuations in the intensity of other bands are apparent.

A quantification of the bands should be performed to be able to state that only FPN1 is altered.

The quantification of the other surface proteins (namely TfR) by cytometry should have also

been performed, to have a more complete picture of iron-related alterations.

Answer: Thank you for your valuable suggestions. The values quantitated by measuring band intensities using the “ImageJ” software were normalized to those of actin. We added this method in the “Western blot” part of revised manuscript. Also, the values of DMT1, TFR1 and TFR2 of Figure 1e, f, i, S1k, l were marked and no differences were observed in statistics analysis. We added these data in the revised manuscript.

Figure 1e

Figure 1f

Figure S1k THP-1

Figure S1l THP-1

Besides, cell surface DMT1, TFR1 and TFR2 were detected by flow cytometry analysis. Consistent with the results obtained from western blot, the surface proteins including TFR or DMT1 remain unchanged after viral infection and IFN- β treatment (Fig. 1h, S1m). We added these data in the revised manuscript.

Figure 1h

Figure S1m

Figure S8g

5. Several passages are very confusing and/or scientifically incorrect.

Line 108 “FPN1 deficiency abolished the decrease in total cellular iron and ferrous levels...”- and line 110 “viruses disrupt iron withholding by suppressing the host FPN1 expression” : besides being questionable as interpretation of the data, these sentences make no sense per se. In the first case, I guess the authors meant “FPN1 deficiency abolished the increase in total cellular iron and ferrous iron levels...”. In the second case, “viruses disrupt iron release, by suppressing the host FPN1 expression”.

Answer: Thank you for your comments and suggestion. As your suggestion, we replaced the first case “FPN1 deficiency abolished the decrease...” with “FPN1 deficiency abolished the increase...”. For the second case, iron withholding is a kind of defense mechanism adopted from host to limit iron acquisition by invading pathogens. For viruses who relying entirely on host materials for replication and packaging, host needs to export iron to extracellular space. So “viruses disrupt iron withholding” means “viruses disrupt iron release”. Because our research mainly discusses the impact of viruses on iron withholding, we did not revise this sentence.

6. Figure 2d- The way the RNA-seq results are presented is not very obvious. A Venn diagram would be more intuitive. The legend of the figure states “Significantly enriched biological processes of the E3 ligase as RNA...”, which is a very confusing way of describing the results. Additionally, the way the RNA-seq method is described is not clear, particularly lines 430-433: “Genes with a fold change (FC) of >4 and an FDR of <0.001 after three treatments with HSV-1, VSV, and SeV were selected for further study. The identified differences were intersected with the E3 gene to obtain E3 with an FC of >4 and an FDR of <0.001. Subsequently, a

schematic diagram was generated.”

Answer: Thank you for your valuable suggestions. Considered that volcanic map could more clearly show the significance of the molecules we have screened out, so we did not replace it with Venn diagram.

Response Fig. 3

The groups which PMs infection with SeV (MOI:1) for 8 h, VSV (MOI:1) for 8 h, or HSV-1 (MOI:10) for 4 h (n = 3) were subjected to RNA-seq analysis, and the differentially upregulated E3 ubiquitin ligases expression was visualized using Venn diagram ($\log_2FC > 2$). We added this description in the legend of revised manuscript.

Scatter plot of RNA-sequencing data displays the mouse PMs upregulated E3 ubiquitin ligases in control or infection with SeV (MOI:1) for 8 h, VSV (MOI:1) for 8 h, or HSV-1 (MOI:10) for 4 h (n = 3). Colors represent fold-change levels above ($\log_2FC > 2$; red) the values in control. (**FDR < 0.001). We rewrote the legend in the revised manuscript.

To determine genes expression changes after virus infection (HSV-1, VSV, and SeV) by RNA-Seq experiment. First, the RNA-seq data were processed by the edgeR package to analyze the

paired differential genes, the p values underwent FDR correction. Among 291 genes with E3 activity, which were chosen as significantly upregulated after viral infection, with log₂FC of >2 and FDR of <0.001 as statistical boundaries. Finally, the significantly upregulated E3 activity genes were visualized using a schematic diagram. We rewrote the method in the revised manuscript.

7. The term “ferrous” is often used instead of “iron”. For example: Line 24 (in the abstract) and line 71 (in the introduction) “ ferrous suppresses type I IFN” should read “ferrous iron (or simply iron) suppresses type I IFN”. Line 93: “accumulation of ferrous in PMs”

Answer: Thank you for your comments and suggestion. We have replaced all the “ferrous” with “ferrous iron” in the revised manuscript.

8. It would be important to have the exact number (or verifiable reference) of the licence(s) obtained to perform all the procedures involving animals.

Answer: Thank you for your comments and suggestion. The number of the licence obtained to perform all the procedures involving animals is ECSBMSSDU2021-2-048. We added this information in the revised manuscript.

Reviewer #6 :I've been asked to comment on the revisions performed to address Reviewer #5's concerns. The responses all appear adequate. The inconsistencies and lack of methodological details noted by the reviewer have been addressed.

Answer: We appreciate very much for your work in reviewing our manuscript. The insightful suggestions and comments help us to greatly improve the manuscript.